# DIFFERENTIABLE LIFTING FOR TOPOLOGICAL NEURAL NETWORKS

**Jorge Luiz Franco**
University of São Paulo, Instituto Curvelo

**Gabriel Duarte**
Federal Institute of Ceará

**Alexander Nikitin**
Aalto University

**Moacir Ponti**
University of São Paulo

**Diego Mesquita**
Getulio Vargas Foundation, $2\delta$ AI

**Amauri H. Souza**
Federal Institute of Ceará, $2\delta$ AI

## ABSTRACT

Topological neural networks (TNNs) enable leveraging high-order structures on graphs (e.g., cycles and cliques) to boost the expressive power of message-passing neural networks. In turn, however, these structures are typically identified *a priori* through an unsupervised graph lifting operation. Notwithstanding, this choice is crucial and may have a drastic impact on a TNN's performance on downstream tasks. To circumvent this issue, we propose $\partial$lift (DiffLift), a general framework for learning graph liftings to hypergraphs and cellular- and simplicial complexes in an end-to-end fashion. In particular, our approach leverages learned vertex-level latent representations to identify and parameterize distributions over candidate higher-order cells for inclusion. This results in a scalable model which can be readily integrated into any TNN. Our experiments show that $\partial$lift outperforms existing lifting methods on multiple benchmarks for graph and node classification across different TNN architectures. Notably, our approach leads to gains of up to 45% over static liftings, including both connectivity- and feature-based ones.

## 1 INTRODUCTION

Topological neural networks (TNNs) (Papillon et al., 2023b; Bodnar et al., 2021a; Verma et al., 2024) have recently emerged as a prominent class of models for learning on topological domains, such as hypergraphs and simplicial complexes, with many researchers arguing they represent the new frontier for relational learning (Papamarkou et al., 2024). Akin to graph neural networks (GNNs) (Scarselli et al., 2009; Gilmer et al., 2017), typical TNNs employ message-passing layers where each element of the input (e.g., nodes or cells) updates its representation (features) based on those of its topological neighbors. Thus, these models generalize convolution-like operations on graphs to higher-order relational objects. Importantly, the primary application of TNNs has been to enhance the capabilities of graph-based models, particularly in terms of expressivity (Bodnar et al., 2021a;b). In this context, the input graphs must first be transformed to the domain on which a TNN operates — a process known as *lifting*.

Lifting methods explore graph connectivity and features to create higher-order relational structures. For instance, *clique lifting* (Bodnar et al., 2021b) produces a simplicial complex by leveraging cliques in the input graph while *cycle lifting* (Hajij et al., 2022) detects cycles to create a cell complex. In general, there are many lifting procedures for each topological domain — c.f. Tab. 3 in Telyatnikov et al. (2025).

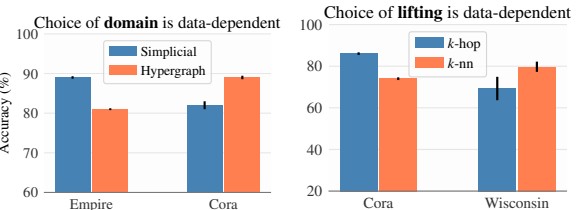

Figure 1: [*Left*] Lifting to different domains can lead to disparate performances. Accuracies taken from the best TNNs in (Telyatnikov et al., 2025). [*Right*] Performances of liftings to the same domain (hypergraph) vary greatly. Values taken from Table 4.

Not surprisingly, the optimal choice of topological domain and lifting procedure for each task is non-obvious, and its impact on performance is highly data-dependent. Figure 1 compares TNNs on different domains, showing opposite behaviors depending on data, even within the same topological domain.

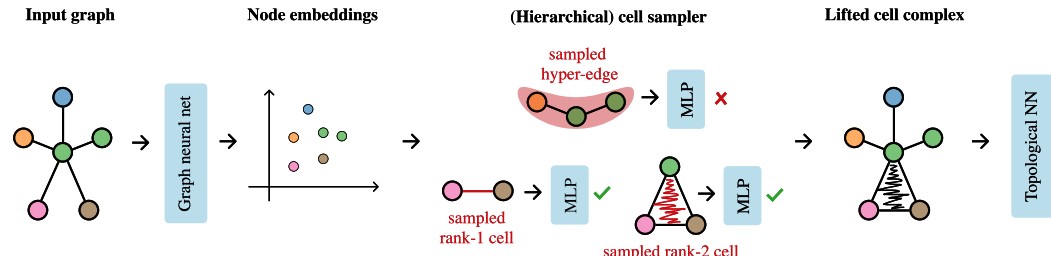

Figure 2: Overview of $\partial$lift. For a given input graph, we first compute node embeddings using GNNs. Then, we use these embeddings to select cells/hyperedges. Cell-level embeddings run through MLPs responsible for returning acceptance probabilities. For hierarchical domains (e.g., cell complexes), cells are generated in increasing dimensionality. From the accepted cells, we form a relational object that is sent to an off-the-shelf TNN for graph/node-level predictions. The model is trained end-to-end.

Strikingly, despite the high impact of the lifting operation on TNNs, most lifting methods are not supervised and thus not informed by the task at hand (Hajij et al., 2022; Telyatnikov et al., 2025), which may lead to suboptimal architectures. To date, differentiable lifting has only been explored in the context of cell complexes (Battiloro et al., 2024).

This work proposes $\partial$lift (DiffLift) – a general, differentiable lifting framework applicable to various domains, including hypergraphs and simplicial/cell complexes. Our method uses a probabilistic approach to sample candidate cells of adaptive sizes. Specifically, we parameterize distributions over cells using node embeddings derived from arbitrary graph models (e.g., GNNs or Graph Transformers (Rampášek et al., 2022)). For each candidate cell, we compute its embedding and use a multilayer perceptron (MLP) to estimate the probability of accepting or rejecting the cell — that is, determining whether it should be included in the output structure. Figure 2 provides a schematic overview of $\partial$lift.

To model the typical hierarchical structure of topological objects, we propose an iterative sampling procedure, where cells are generated in increasing order of dimensionality: samples of dimension $i$ are used to inform the sampling of dimension $(i + 1)$-cells. Notably, our approach generalizes across multiple topological domains and can be seamlessly integrated into standard TNN pipelines.

We evaluate $\partial$lift on 12 datasets spanning graph and node classification tasks using four different TNN models. Our results show that $\partial$lift consistently outperforms unsupervised lifting methods in nearly all graph-level classification benchmarks — achieving superior performance in 22 out of 24 experiments, often by a substantial margin. These gains are robust across all TNN architectures. For example, when using CW Networks (Bodnar et al., 2021a), $\partial$lift yields performance gains of up to 45%. For node classification, $\partial$lift achieves competitive performance relative to static lifting methods and outperforms DCM (Battiloro et al., 2024) (a differentiable lifting baseline) overall. Additionally, we analyze the sensitivity of $\partial$lift to the choice of its GNN component, highlighting that while this choice often impacts the overall performance, our design is robust and produces strong empirical results even when adopting simple GNNs (e.g., graph isomorphism networks (Xu et al., 2019)).

## 2 BACKGROUND

This section overviews the main types of relational structures and respective neighborhood notions, message-passing networks for relational data, and graph lifting methods. In the following, we assume readers are familiar with basic notions in topology; see (Munkres, 2000) for reference.

**Graphs and hypergraphs.** We denote an *undirected graph* as a tuple $G = (V, E)$ where $V$ is a set of vertices (or nodes) and $E$ is a set of unordered vertex pairs, i.e., edges. The set of neighbors of a node $v$ in $G$ is denoted by $\mathcal{N}^G(v) = \{u \in V : \{v, u\} \in E\}$. Hypergraphs generalize graphs by allowing edges to connect multiple nodes. Formally, a *hypergraph* on a nonempty set $V$ is a pair $(V, K)$, where $K \subseteq 2^V \setminus \emptyset$ and its elements are called hyperedges.

**Simplicial complexes** are topological spaces comprised of simple mathematical objects called simplices (points (0-simplices), line segments (1-simplices), triangles (2-simplices), and their higher-dimensional analogues). In particular, an *abstract simplicial complex* (ASC) over a vertex

Graph 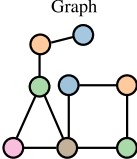 Simplicial Complex 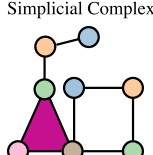 Cell Complex 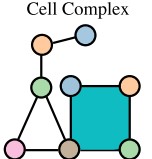 Hypergraph 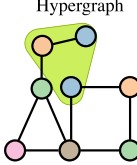 Combinatorial Complex 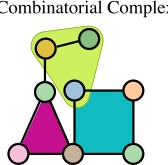

Figure 3: Examples of topological domains.

set $V$ is a set $K$ of subsets of $V$ (the *simplices*) such that, for every $\sigma \in K$ and every non-empty $\tau \subset \sigma$, we have that $\tau \in K$. Thus, we can define ASCs as a family of subsets $K \subseteq 2^V$ of $V$ that is closed under taking subsets. The dimension of a simplex is equal to its cardinality minus 1, and the dimension of an ASC is the maximal dimension of its simplices. We say $\tau$ is on the boundary of a simplex $\sigma$, denoted by $\tau \prec \sigma$, iff $\tau \subset \sigma$ and there is no $\delta$ such that $\tau \subset \delta \subset \sigma$, i.e., $\prec$ defines the boundary relation of $K$. We note that undirected graphs correspond to 1-dimensional ASCs.

**Cell complexes.** A *regular cell complex* (Hansen and Ghrist, 2019) is a topological space $X$ with a partition $\{X_\sigma\}_{\sigma \in P_X}$ of subspaces $X_\sigma$ of $X$ called *cells* such that

1. For each $x \in X$, there is an open neighborhood of $x$ that intersects finitely many cells;
2. For all $\sigma, \tau \in P_X$, $X_\tau \cap \overline{X_\sigma} \neq \emptyset$ only if $X_\tau \subseteq \overline{X_\sigma}$, where $\overline{X_\sigma}$ denotes the closure of $X_\sigma$ (the intersection of all closed sets containing $X_\sigma$);
3. Every cell $X_\sigma$ is homeomorphic to $\mathbb{R}^{D_\sigma}$ for some $D_\sigma$ which we call $X_\sigma$'s dimension;
4. For all $\sigma \in P_X$, there is a homeomorphism $\phi$ of a closed ball in $\mathbb{R}^{D_\sigma}$ to $\overline{X_\sigma}$ such that the restriction of $\phi$ to the interior of the ball is a homeomorphism onto $X_\sigma$.

Importantly, the conditions (2) and (4) impose a poset structure $\tau \leq \sigma \iff X_\tau \subseteq \overline{X_\sigma}$ which fully characterizes the topology of the underlying cell complex $X$. This topological information can be described by the *boundary relation* $\prec$ between two cells: $\sigma \prec \tau$ iff $\sigma < \tau$ and there is no cell $\delta$ such that $\sigma < \delta < \tau$, where $<$ denotes the strict version of the partial order $\leq$ above. We note that the class of cell complexes subsumes simplicial complexes. For more details on cell complexes, we refer to Hatcher (2002); Bodnar et al. (2021a).

**Combinatorial complexes.** A *combinatorial complex* (CC) (Hajij et al., 2022) is a tuple $(V, K, \mathrm{rk})$ where $V$ is a finite set, $K \subseteq 2^V \setminus \emptyset$ comprises a set of cells, and $\mathrm{rk} : K \to \mathbb{Z}_{\geq 0}$ is a ranking function s.t.

1. For all $v \in V$, $\{v\} \in K$;
2. For all $\sigma, \sigma' \in K, \sigma \subseteq \sigma' \implies \mathrm{rk}(\sigma) \leq \mathrm{rk}(\sigma')$.

The idea of CCs is to generalize hierarchical structures (e.g., simplicial complexes) by imposing mild relationships between cells via ranking functions — CCs only require the order-preserving property in condition (2) — while being flexible to accommodate non-hierarchical structures such as hypergraphs. Figure 3 depicts the most popular relational structures in topological deep learning.

**Neighborhood structures.** We can exploit boundary relations (or rank functions) to specify local neighbors for each cell. In particular, Bodnar et al. (2021b) introduce four neighborhood structures:

- Boundary and co-boundary: $\mathcal{N}_B(\sigma) = \{\tau : \tau \prec \sigma\}$ and $\mathcal{N}_C(\sigma) = \{\tau : \sigma \prec \tau\}$, respectively
- Upper/lower adjacency: $\mathcal{N}_\uparrow(\sigma) = \{\tau : \exists \delta \text{ st } \tau \prec \delta, \sigma \prec \delta\}$ and $\mathcal{N}_\downarrow(\sigma) = \{\tau : \exists \delta \text{ st } \delta \prec \tau, \delta \prec \sigma\}$

Analogs of these neighborhoods can also be obtained via ranking functions (Hajij et al., 2022).

**Features / signals.** In this work, we consider relational structures equipped with features. Let $K$ be a set of cells or hyperedges of a relational domain. Its attributed counterpart is a tuple $(K, x)$ where $x : K \to \mathbb{R}^d$ assigns a feature vector $x(\sigma)$ to each cell $\sigma$. Hereafter, we denote the features of $\sigma$ by $x_\sigma$.

**Topological neural networks (TNNs).** Most TNNs use message-passing mechanisms to obtain cell-level representations (Papillon et al., 2023b). In particular, let $\mathcal{N}_i$ be a finite sequence of neighborhood structures, $\mathcal{N}_C(\sigma, \tau) = \mathcal{N}_C(\sigma) \cap \mathcal{N}_C(\tau)$, and $\mathcal{N}_B(\sigma, \tau) = \mathcal{N}_B(\sigma) \cap \mathcal{N}_B(\tau)$. In its general form,

starting from $h_\sigma^0 = x_\sigma$ for all $\sigma$, a message-passing TNN (Bodnar et al., 2021a) recursively computes

$$m_{i,\sigma}^\ell = \begin{cases} \{\!\{\phi_{\ell,i}(h_\tau^\ell, h_\sigma^\ell, h_\delta^\ell) : \tau \in \mathcal{N}_i(\sigma), \delta \in \mathcal{N}_B(\sigma,\tau)\}\!\}, & \text{if } \mathcal{N}_i = \mathcal{N}_\downarrow \\ \{\!\{\phi_{\ell,i}(h_\tau^\ell, h_\sigma^\ell, h_\delta^\ell) : \tau \in \mathcal{N}_i(\sigma), \delta \in \mathcal{N}_C(\sigma,\tau)\}\!\}, & \text{if } \mathcal{N}_i = \mathcal{N}_\uparrow \\ \{\!\{\phi_{\ell,i}(h_\tau^\ell, h_\sigma^\ell) : \tau \in \mathcal{N}_i(\sigma)\}\!\}, & \text{otherwise.} \end{cases} \tag{1}$$

$$h_\sigma^{\ell+1} = \varphi\left(h_\sigma^\ell, \bigotimes_i \mathrm{Agg}_\ell\left(m_{i,\sigma}^\ell\right)\right) \tag{2}$$

where $h_\sigma^\ell$ is the embedding of $\sigma$ at layer $\ell$, $\bigotimes$ and $\mathrm{Agg}_\ell$ are inter- and intra-neighborhood aggregation functions, respectively, and $\varphi$ is an update function (e.g., MLP).

**Graph lifting.** A *graph lifting* is a map lift : $\mathbb{G} \to \mathbb{T}$ from the space of attributed graphs, $\mathbb{G}$, to a target domain, $\mathbb{T}$, such that $G \cong_\mathbb{G} G' \implies \mathrm{lift}(G) \cong_\mathbb{T} \mathrm{lift}(G')$, where $\cong_\mathbb{T}$ denotes the *isomorphism* relation in domain $\mathbb{T}$. One of the most widely used methods for lifting graphs to cell complexes is *cycle lifting* (Bodnar et al., 2021a). This is a static (non-learnable) approach that constructs 2-cells by identifying basic cycles (elements of a cycle basis) or chordless cycles (Bodnar et al., 2021a) in input graphs. Specifically, the vertices involved in a basic cycle are grouped to form a 2-cell in the resulting cell complex. A cycle basis of a graph $G$ is a minimal set of cycles such that any other cycle in $G$ can be expressed as a modulo-2 sum of cycles from this set.

## 3 DIFFERENTIABLE LIFTING

In this section, we introduce $\partial$lift (read DiffLift), a general framework for learning graph lifting functions. Section 3.1 provides an iterative description of our method, allowing for learning structures of increasingly higher order — when the target domain is hierarchically structured. Section 3.2 and Section 3.3 instantiate $\partial$lift for graph-to-hypergraph and graph-to-cell-complex liftings, respectively. Moreover, we formulate our approach for simplicial- and combinatorial complexes in Appendix A.

### 3.1 GENERAL FORMULATION

Lifting consists of determining which higher-order cells should be added to an input graph $G$, satisfying the constraints of the target domain. To do so, we propose the following recipe.

---

**$\partial$lift : general recipe for differentiable graph liftings**

**Input**: Attributed graph $G = (V, E, x)$, target domain $\mathbb{T}$, and maximum dimension $D_{\max}$.

**Step 1: Compute node embeddings.** Use an arbitrary GNN to compute a vector representation (embedding) $z_v$ for each node $v \in V$. This GNN component can be either a pre-trained model or learned end-to-end. Set the current domain dimension to $D = 1$.

**Step 2: Elicit candidate cells.** Given the node embeddings $\{z_v\}_{v \in V}$, define a set of candidate cells $\mathcal{C} \subseteq 2^V$ of dimension $D$. For each cell $C \in \mathcal{C}$, compute an embedding $z_C = \bigoplus_{v \in C} z_v$, where $\bigoplus$ is an arbitrary permutation-invariant aggregation function. Note that the exact procedure for defining candidate cells depends on the target domain $\mathbb{T}$, as candidates must respect possible hierarchical constraints.

**Step 3: Accept/reject candidate cells.** Apply a neural network $\phi$ (e.g., an MLP) that defines an acceptance probability $\phi(z_C)$ for each candidate cell $C$. Finally, draw a sample $y_C$ from a Bernoulli distribution with parameter $\phi(z_C)$ indicating whether cell $C$ is accepted or not. The resulting domain is then given by $V \cup E \cup \{C \in \mathcal{C} : y_C = 1 \text{ with } y_C \sim \mathrm{Ber}(\phi(z_C))\}$.

**Step 4: Termination check.** If $D = D_{\max}$, halt; otherwise, $D \leftarrow D + 1$ and return to Step 2.

---

Importantly, $\partial$lift is learned in an end-to-end fashion, using the straight-through estimator (Bengio et al., 2013) to propagate gradients through samples at Step 3. For hypergraphs, we assume hyperedges have dimension one, causing $\partial$lift to stop once it reaches Step 4. We note that Steps 2 and 3 are the only domain-dependent ingredients of our algorithm. Next, we explain how these steps can be adapted to specific domains.

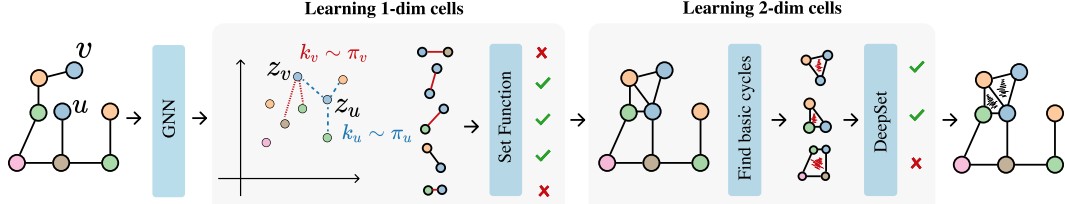

Figure 4: Two iterations of $\partial$lift for cell complexes. At the first iteration, we leverage each node $u$'s GNN embedding $z_u$ to delineate candidate 1-dim cells. Specifically, we consider cell-equivalent of edges linking $u$ to each of its $k_u$ NNs in embedding space, where $k_u$ is a random variable parameterized by $z_u$. We use a set function over the embedding of nodes within each cell to compute their acceptance probabilities. At the second iteration onwards, we use cycle lifting in our augmented cell complex to elicit candidate cells, whose acceptance probabilities are computed similarly to the first step.

## 3.2 GRAPH-TO-HYPERGRAPH LIFTING

$\Rightarrow$[Step 2] For notational convenience, suppose we wish to learn up to one hyperedge per node. For each node $v$, we define a candidate hyperedge $C(v)$ using the $k_v$ nearest neighbors of $v$ in the embedding space:

$$C(v) = \{S \subset V : |S| = k_v \text{ and } w \notin S \implies \text{dist}(z_w, z_v) \geq \max_{u \in S} \text{dist}(z_u, z_v)\}, \tag{3}$$

where $\text{dist}(\cdot, \cdot)$ denotes a dissimilarity metric. Here, we consider the Euclidean distance.

To allow for adaptive hyperedge sizes, we sample $k_v$ according to a probability distribution parameterized by (a function of) $v$'s embedding $z_v$. More specifically, we define the $(k_{\max} - k_{\min} + 1)$-dimensional probability vector $\pi_v \propto \exp \circ \text{MLP}(z_v)$ and draw $k_v \sim \text{Categorical}(\pi_v)$, where $k_{\min}$ and $k_{\max}$ are lower- and upper-bounds on $k_v$.

$\Rightarrow$[Step 3] We define the probability of acceptance (i.e., of $b_v = 1$) for $C(v)$ as a function of the (multiset of embeddings of) nodes in $C(v)$. More specifically, we define $b_v$ as

$$b_v \sim \text{Ber}(\Psi(\{\!\{z_u : u \in C(v)\}\!\})), \tag{4}$$

where $\Psi$ is learned and maps from multisets (i.e., $\Psi$ is order-invariant) of elements in $\mathbb{R}^d$ to $(0, 1)$.

**Feature lifting.** Each accepted hyperedge $C(v)$ receives a feature vector $x_{C(v)}$ computed as a multiset operation over $\{\!\{x_u : u \in C(v)\}\!\}$. Specifically, we employ a *scaled sum projection*:

$$x_{C(v)} = \frac{1}{k_v} \sum_{u \in C(v)} x_u, \quad \forall v \text{ such that } b_v = 1. \tag{5}$$

## 3.3 GRAPH-TO-CELL-COMPLEX LIFTING

For computational reasons, we split the lifting procedure for cell complexes into two cases. We provide an overview of our proposed graph-to-cell-complex lifting in Figure 4.

**Case $D = 1$: Learning edges**

$\Rightarrow$[Step 2] Similarly to Step 2 of graph-to-hypergraph lifting, for each node $v$, we sample a neighborhood size $k_v$ and define a set $C(v) \subseteq V \setminus \{v\}$ containing the nodes associated with the $k_v$ nearest neighbors of $v$ in the embedding space, excluding $v$ itself.

$\Rightarrow$[Step 3] Next, we construct candidate edges (1-cells) by considering each pair $(v, v')$ and define their probability of acceptance (i.e., of $b_{v,v'} = 1$) as a function of the embeddings of $v$ and $v'$. Specifically, we set $b_{v,v'} \sim \text{Ber}(\Psi(\{\!\{z_v, z_{v'}\}\!\}))$, where $\Psi$ is an order-invariant function.

At end of this iteration, the obtained cell complex is given by:

$$K^1 = V \cup E \cup \{\{v, v'\} : v \in V, v' \in C(v) \text{ with } b_{v,v'} = 1\}. \tag{6}$$

Regarding *feature lifting*, we apply scaled sum projection, identically to the hypergraph case.

**Case $D \geq 2$: Learning $D$-cells**

To select candidate cells of arbitrary dimension, we need the notion of $n$-cycles of a cell complex. Let $C_n(K)$ denote the $n$-chains of the cell complex $K$ equipped with $\mathbb{Z}/2\mathbb{Z}$-vector space structure. Also let $\partial_n : C_n(K) \to C_{n-1}(K)$ be the boundary linear map on $K$. Then, the $n$-cycles of $K$ are given by $Z_n(K) = \ker(\partial_n)$. We provide further details in the supplementary material.

$\Rightarrow$[Step 2] Let $K^{D-1}$ be the cell complex at the end of iteration $D-1$. We select a basis for $(D-1)$-set of cycles $Z_{D-1}(K^{D-1})$ in $K^{D-1}$ to serve as candidate cells. Recall a basis for cycles is a minimal collection of cycles such that any cycle can be written as a modulo-2 sum of cycles in the basis. We note that we can also employ chordless cycles to define the 2-cells, as in (Bodnar et al., 2021a).

$\Rightarrow$[Step 3] Let $\mathcal{C}$ be the set of candidate $(D-1)$-cycles from Step 2. We define the probability of accepting $C \in \mathcal{C}$ (i.e., setting $b_C = 1$) using a DeepSet model (Zaheer et al., 2017) over the multiset of embeddings $\{\!\{z_v\}\!\}_{v \in C}$ of all nodes $v$ in $C$. The output complex at this iteration is then

$$K^D = K^{D-1} \cup \{C \in \mathcal{C} : b_C = 1\}. \tag{7}$$

For simplicity, again, the features of $D$-cells are obtained via sum projection lifting.

**Remark 1** *Despite the generality of $\partial$lift, in the experiments we only consider 2-dimensional cell complexes ($D_{max} = 2$) and use the algorithm in (Paton, 1969) (available at the toolbox NetworkX (Hagberg et al., 2008)) to identify basic 1-cycles in graphs. This is mainly due to the fact that current implementations of TNNs for cell complexes only support 2-dimensional objects — for instance, see TopoBench (Telyatnikov et al., 2025).*

**Remark 2** *We can obtain a deterministic version of $\partial$lift using a probability threshold, i.e., we simply set $b_C = \mathbb{1}[\Psi(\cdot) > \gamma]$ with, e.g., $\gamma = 0.5$, for all candidate cells $C$.*

Experiments regarding the deterministic version can be found in the Appendix E.2.

## 4 RELATED WORKS

**Topological deep learning.** Traditional graph deep learning methods are limited to modeling only pairwise interactions, making them unsuitable for capturing higher-order dependencies involving multiple nodes (Hajij et al., 2022; Papillon et al., 2023a). To address this limitation, a variety of deep topological learning methods have been developed for hypergraphs (Bai et al., 2021; Yadati et al., 2019), simplicial complexes (Hajij et al., 2021; Goh et al., 2022; Maggs et al., 2024; Yang et al., 2022), and cell complexes (Hajij et al., 2022; 2020), the works that are based on the topological signal processing field (Barbarossa and Sardellitti, 2020; Schaub et al., 2021; Roddenberry et al., 2022; Sardellitti et al., 2021). Papillon et al. (2025) also use GNNs to enhance TDL, where the lifted topological domain is transformed into augmented Hasse graphs. These methods have demonstrated their effectiveness across several practical applications, including action recognition (Wang et al., 2022; Hao et al., 2021), bioinformatics (Liu et al., 2022), and neuroscience (Wang et al., 2023).

**Liftings to topological domains.** Most relational datasets and benchmarks are defined on discrete structures such as graphs. To apply topological deep learning methods to these datasets, a transformation process known as lifting is required, which maps discrete data into topological domains (Telyatnikov et al., 2025; Hajij et al., 2022; Bernárdez et al., 2024). This lifting process can be either predefined – e.g., based on structural features like node proximity or the presence of cycles – or learned directly from the data (Battiloro et al., 2024; Kazi et al., 2022). Graph structure learning methods (Qian et al., 2024; Kazi et al., 2022; Franceschi et al., 2019; Topping et al., 2022; Sun et al., 2023; Chen et al., 2020; Jin et al., 2020) are closely related to the graph lifting literature and can be interpreted as instances of graph lifting to graph domain. Our approach represents the most general form of learnable lifting proposed so far and empirically outperforms the aforementioned methods in many benchmarks.

**Static liftings.** To the best of our knowledge, Bodnar et al. (2021b;a) were the first to combine static liftings and high-order message passing, focusing on simplicial- and cell complexes. These static liftings embed a graph into a topological domain by, e.g., aggregating each node's n-hop neighborhood or by tracing its cycles. The repertoire of static liftings was later broadened by

the ICML TDL challenge (Bernárdez et al., 2024), which added methods based on kNN, Voronoi decompositions, and random walks. Our work proposes a more flexible, data-driven approach to defining liftings, which offers benefits across a range of tasks.

## 5 EXPERIMENTS

In this section, we evaluate $\partial$lift on two complementary tasks: graph classification and node classification. We compare it against broadly used lifting schemes for both hypergraphs and cell complexes. We also report results across different TNNs within each of these domains. We run experiments using PyTorch (Paszke et al., 2017) and PyTorch Geometric (Fey and Lenssen, 2019); our code is anonymously available at `https://github.com/JorgeLuizFranco/difflifting`.

### 5.1 GRAPH CLASSIFICATION

**Datasets.** We evaluate model performance on six widely used graph-level benchmark datasets for molecular property prediction: NCI1, NCI109, MUTAG, MOLHIV, PROTEINS, and ZINC (Kersting et al., 2016; Dwivedi et al., 2023; Hu et al., 2020). These datasets are standard benchmarks in the literature for assessing the effectiveness of graph-based models (Dwivedi et al., 2023; Telyatnikov et al., 2025). All tasks are binary classification problems, with the exception of ZINC, which is a regression task. We provide more details regarding datasets in the Appendix.

**Baselines.** We compare $\partial$lift with four existing graph lifting methods: cycle lifting, $k$-hop lifting, $k$-nearest-neighbor ($k$-NN) lifting, and kernel lifting. Among these, cycle lifting is the most widely adopted strategy for graph-to-cell-complex liftings and has become the *de facto* standard in most of TNNs operating on cell complexes (Telyatnikov et al., 2025). Similarly, $k$-hop lifting is the predominant approach for constructing hypergraphs from graphs and is often the sole method considered in recent benchmarks such as (Telyatnikov et al., 2025). We also consider $k$-NN lifting as it shares similarities with our approach due to the use of $k$-NN. Finally, we consider kernel lifting, one of the most successful approaches in the ICML TDL challenge (Bernárdez et al., 2024). Notably, our choice of baselines covers lifting methods based on connectivity (cycle and $k$-hop liftings), features ($k$-NN lifting), and both connectivity and features (kernel lifting). We provide formulations for the baseline liftings in Appendix B. We consider the following TNNs: CWN, CIN (Bodnar et al., 2021b) and CXN (Hajij et al., 2020) for cell complexes, and UniGCNII and UniGIN (Huang and Yang, 2021) for hypergraphs.

**Evaluation setup.** For ZINC and MOLHIV, we use the publicly available train/val/test data splits; for the remaining datasets, we use a random 80/10/10% split. We optimize the hyper-parameters of the lifting methods and take the optimal hyperparameter values from (Telyatnikov et al., 2025) whenever available; otherwise, we select optimal values based on the optimal results using cycle or $k$-hop lifting. We provide further details on the choice of hyperparameters and model selection in the supplementary material. We compute the mean and standard deviation of the performance metrics (MAE $\downarrow$ for ZINC, AUC $\uparrow$ for MOLHIV, and accuracy $\uparrow$ for all other datasets) over three independent runs.

**Results.** Table 1 shows that $\partial$lift is the best-performing lifting method in over 90% of the TNN/dataset combinations, both for cell complexes and hypergraphs. Notably, $\partial$lift resulted in an improvement in average accuracy of up to $\approx 45\%$ compared to static liftings using the same TNN. For CWN, CIN, and UniGCNII, our method outperforms static liftings on all datasets. On NCI109 and ZINC, $\partial$lift is consistently better than the static liftings across all TNN backbones. We also observe that TNNs perform better than GNNs in this setup with $\partial$lift. Overall, these results validate the effectiveness of $\partial$lift.

**Impact of GNN choice on performance.** We aim to assess how sensitive our approach is to the choice of GNN. We report results using GIN (Xu et al., 2019) and GPS (Rampášek et al., 2022).

Table 2 indicates that choosing GNNs that are able to generate richer and more informative latent node representations leads to better results in $\partial$lift. In particular, GPS performs better than GIN in most datasets. A possible explanation for this observation is the greater expressivity of GPS, which benefits from the incorporation of positional encodings. Notably, on cell complexes and ZINC dataset, GPS allows reducing the MAE from 0.46 to 0.17.

Table 1: Graph classification: $\partial$lift vs static liftings. We denote the best-performing model for each dataset/TNN in bold. For any fixed TNN and dataset, $\partial$lift is better than static liftings in 90% of cases, offering a performance improvement of up to 45%.

| Domain | TNN | Lifting | NCI1↑ | NCI109 ↑ | MOLHIV↑ | MUTAG↑ | Proteins↑ | ZINC↓ |
|---|---|---|---|---|---|---|---|---|
| Graph | GCN | - | $74.45_{\pm1.05}$ | $76.46_{\pm1.03}$ | $74.99_{\pm1.09}$ | $64.91_{\pm4.96}$ | $70.18_{\pm1.35}$ | $0.64_{\pm0.04}$ |
| | GIN | - | $76.89_{\pm1.75}$ | $76.90_{\pm0.80}$ | $70.76_{\pm2.46}$ | $80.70_{\pm2.48}$ | $72.50_{\pm2.31}$ | $0.59_{\pm0.03}$ |
| Cellular | CWN | Cycle | $76.93_{\pm1.18}$ | $76.71_{\pm1.34}$ | $70.15_{\pm3.98}$ | $66.67_{\pm12.41}$ | $69.05_{\pm2.95}$ | $0.46_{\pm0.01}$ |
| | | $\partial$lift | $\mathbf{79.81}_{\pm0.40}$ | $\mathbf{80.55}_{\pm0.50}$ | $\mathbf{75.37}_{\pm0.80}$ | $\mathbf{85.96}_{\pm4.96}$ | $\mathbf{70.54}_{\pm3.34}$ | $\mathbf{0.17}_{\pm0.00}$ |
| | CXN | Cycle | $72.02_{\pm1.69}$ | $75.01_{\pm0.62}$ | $69.17_{\pm1.20}$ | $61.40_{\pm2.48}$ | $\mathbf{70.83}_{\pm1.52}$ | $0.79_{\pm0.02}$ |
| | | $\partial$lift | $\mathbf{82.08}_{\pm1.50}$ | $\mathbf{82.57}_{\pm0.40}$ | $\mathbf{74.83}_{\pm1.96}$ | $\mathbf{84.21}_{\pm4.30}$ | $69.94_{\pm2.10}$ | $\mathbf{0.17}_{\pm0.01}$ |
| | CIN | Cycle | $75.91_{\pm1.11}$ | $76.11_{\pm1.09}$ | $68.46_{\pm2.16}$ | $66.96_{\pm1.46}$ | $67.86_{\pm0.89}$ | $0.42_{\pm0.01}$ |
| | | $\partial$lift | $\mathbf{79.59}_{\pm1.50}$ | $\mathbf{81.06}_{\pm0.40}$ | $\mathbf{72.37}_{\pm1.65}$ | $\mathbf{88.72}_{\pm4.30}$ | $\mathbf{72.43}_{\pm2.10}$ | $\mathbf{0.20}_{\pm0.01}$ |
| Hypergraph | UniGCN2 | $k$-hop | $72.70_{\pm0.52}$ | $72.01_{\pm1.55}$ | $50.72_{\pm1.06}$ | $61.40_{\pm2.48}$ | $72.92_{\pm1.11}$ | $0.66_{\pm0.02}$ |
| | | $k$-NN | $71.78_{\pm0.20}$ | $68.60_{\pm0.93}$ | $57.73_{\pm6.84}$ | $64.91_{\pm2.48}$ | $73.51_{\pm0.42}$ | $1.10_{\pm0.01}$ |
| | | kernel | $73.80_{\pm0.94}$ | $72.64_{\pm0.40}$ | $57.07_{\pm10.32}$ | $63.16_{\pm8.59}$ | $73.21_{\pm0.73}$ | $0.79_{\pm0.02}$ |
| | | $\partial$lift | $\mathbf{77.45}_{\pm1.88}$ | $\mathbf{75.30}_{\pm1.10}$ | $\mathbf{69.32}_{\pm1.62}$ | $\mathbf{89.47}_{\pm4.30}$ | $\mathbf{73.51}_{\pm0.84}$ | $\mathbf{0.56}_{\pm0.03}$ |
| | UniGIN | $k$-hop | $65.50_{\pm1.99}$ | $66.97_{\pm7.25}$ | $63.49_{\pm9.55}$ | $64.91_{\pm2.48}$ | $71.43_{\pm0.73}$ | $1.15_{\pm0.01}$ |
| | | $k$-NN | $\mathbf{72.83}_{\pm1.09}$ | $70.14_{\pm1.48}$ | $52.34_{\pm3.21}$ | $59.65_{\pm4.96}$ | $72.62_{\pm1.52}$ | $1.10_{\pm0.02}$ |
| | | kernel | $60.50_{\pm1.26}$ | $66.59_{\pm1.49}$ | $49.60_{\pm0.07}$ | $57.89_{\pm4.30}$ | $66.67_{\pm1.83}$ | $1.45_{\pm0.02}$ |
| | | $\partial$lift | $64.88_{\pm1.09}$ | $\mathbf{79.74}_{\pm0.23}$ | $\mathbf{72.04}_{\pm0.88}$ | $\mathbf{66.67}_{\pm6.56}$ | $\mathbf{73.81}_{\pm1.52}$ | $\mathbf{0.92}_{\pm0.05}$ |

Table 2: Effect of GNN backbone on the performance of $\partial$lift. The results suggest that the expressive power of backbone GNNs have a direct impact in $\partial$lift's performance. Except for MOLHIV and NCI1, GPS leads to better performance than GIN overall.

| TNN | GNN | NCI1↑ | NCI109 ↑ | MOLHIV ↑ | MUTAG↑ | Proteins↑ | ZINC↓ |
|---|---|---|---|---|---|---|---|
| CWN | GPS | $79.81_{\pm0.40}$ | $\mathbf{80.55}_{\pm0.50}$ | $64.31_{\pm5.32}$ | $\mathbf{87.72}_{\pm2.48}$ | $70.54_{\pm3.34}$ | $\mathbf{0.17}_{\pm0.00}$ |
| | GIN | $\mathbf{81.59}_{\pm0.80}$ | $78.69_{\pm1.43}$ | $\mathbf{75.37}_{\pm0.80}$ | $82.46_{\pm2.48}$ | $\mathbf{71.13}_{\pm2.76}$ | $0.46_{\pm0.00}$ |
| CXN | GPS | $79.97_{\pm0.61}$ | $\mathbf{82.57}_{\pm0.40}$ | $65.58_{\pm4.42}$ | $\mathbf{84.21}_{\pm4.30}$ | $\mathbf{69.94}_{\pm2.10}$ | $\mathbf{0.17}_{\pm0.01}$ |
| | GIN | $\mathbf{81.35}_{\pm2.29}$ | $79.98_{\pm1.01}$ | $\mathbf{72.25}_{\pm3.23}$ | $77.19_{\pm2.48}$ | $67.86_{\pm2.19}$ | $0.43_{\pm0.01}$ |
| UniGCN2 | GPS | $\mathbf{78.67}_{\pm1.46}$ | $74.50_{\pm1.16}$ | $68.22_{\pm2.38}$ | $\mathbf{89.47}_{\pm4.30}$ | $\mathbf{73.81}_{\pm0.42}$ | $\mathbf{0.56}_{\pm0.03}$ |
| | GIN | $75.38_{\pm1.39}$ | $\mathbf{74.98}_{\pm1.12}$ | $\mathbf{68.73}_{\pm2.05}$ | $64.91_{\pm6.56}$ | $73.51_{\pm0.84}$ | $0.63_{\pm0.01}$ |
| UniGIN | GPS | $\mathbf{66.42}_{\pm1.79}$ | $\mathbf{79.74}_{\pm0.23}$ | $68.32_{\pm3.12}$ | $66.67_{\pm6.56}$ | $\mathbf{72.32}_{\pm0.73}$ | $\mathbf{1.01}_{\pm0.05}$ |
| | GIN | $64.40_{\pm0.41}$ | $78.53_{\pm0.69}$ | $\mathbf{68.86}_{\pm3.05}$ | $\mathbf{70.18}_{\pm6.56}$ | $72.02_{\pm3.74}$ | $1.12_{\pm0.26}$ |

## 5.2 NODE CLASSIFICATION

**Datasets.** For node classification, we evaluate $\partial$lift on four datasets: Cora, Citeseer (Yang et al., 2016), Texas, and Wisconsin (Rozemberczki et al., 2021). Within these datasets, two are knowingly homophilic (Cora and Citeseer) and two are heterophilic datasets (Texas and Wisconsin). Dataset statistics can be found in Appendix D.

**Baselines.** We also compare our method (for cell domains) against the learnable approach in (Battiloro et al., 2024), called Differentiable Cell Complex Module (DCM), which was originally evaluated on node classification tasks. To do so, we consider $\partial$lift combined with CWN and TopoTune (Papillon et al., 2025). We also include results of them with cycle lifting. We consider the same hypergraph TNN baselines as in Section 5.1.

**Evaluation setup.** For all datasets, we use random train/val/test data split with 60/20/20% split. Similarly to the experiments for graph classification, we optimize the hyper-parameters of the lifting methods and take the optimal TNN hyperparameters from Telyatnikov et al. (2025) when available. Otherwise, we choose them to maximize the validation accuracy using $k$-hop lifting. For more details, please refer to the supplementary material. We report the average accuracy and standard deviation over three independent runs.

**Results.** Table 3 compares $\partial$lift against DCM and cycle lifting. Notably, $\partial$lift is the best-performing method in all datasets except for Wisconsin, in which it achieves the second-best performance. It is

also worth mentioning that $\partial$lift (with either CWN or TopoTune) outperforms DCM for all datasets, sometimes by a large margin — c.f., Texas and Cora.

Table 3: Comparison of DCM and $\partial$lift on node classification. $\partial$lift achieves the highest average performance across all datasets (rhs) and significantly outperforms DCM on heterophilic datasets.

| TNN | Lifting | Cora | Citeseer | Texas | Wisconsin | Avg |
|---|---|---|---|---|---|---|
| GCN | - | $85.64 \pm 0.51$ | $70.43 \pm 0.71$ | $58.91 \pm 0.76$ | $49.14 \pm 0.66$ | 66.03 |
| GAT | - | $86.17 \pm 0.33$ | $73.82 \pm 0.45$ | $58.38 \pm 1.05$ | $49.41 \pm 0.95$ | 66.95 |
| GIN | - | $85.50 \pm 0.54$ | $72.20 \pm 0.60$ | $59.10 \pm 0.80$ | $48.50 \pm 0.70$ | 66.33 |
| DCM | - | $80.73 \pm 0.33$ | $77.90 \pm 0.80$ | $56.76 \pm 6.62$ | $73.86 \pm 1.85$ | 72.31 |
| CWN | Cycle | $74.80 \pm 0.08$ | $75.83 \pm 0.90$ | $63.06 \pm 7.75$ | $\mathbf{80.39} \pm 4.24$ | 73.52 |
| CWN | $\partial$Lift | $80.17 \pm 1.59$ | $72.83 \pm 2.15$ | $\mathbf{80.18} \pm 3.37$ | $77.78 \pm 3.70$ | 77.74 |
| TopoTune | Cycle | $69.03 \pm 0.88$ | $72.90 \pm 0.85$ | $71.56 \pm 1.27$ | $70.16 \pm 1.85$ | 70.91 |
| TopoTune | $\partial$Lift | $\mathbf{86.82} \pm 0.75$ | $\mathbf{78.23} \pm 1.08$ | $72.97 \pm 0.00$ | $65.36 \pm 2.45$ | $\mathbf{75.84}$ |

Table 4 reports results of lifting methods for hypergraph neural networks. Compared to $k$-hop, our approach is better on heterophilic datasets but worse on homophilic ones for UniGCN2. Additionally, Table 4 shows that $\partial$lift leads to better average accuracy than other static liftings.

Table 4: Comparison of $\partial$lift and static lifting baselines for hypergraphs on node classification. Our method outperforms all static liftings on average across the selected node classification datasets.

| TNN | Lifting | Cora | Citeseer | Texas | Wisconsin | Avg |
|---|---|---|---|---|---|---|
| UniGCN2 | $k$-hop | $\mathbf{86.03} \pm 0.63$ | $\mathbf{78.40} \pm 0.36$ | $66.67 \pm 6.74$ | $69.28 \pm 5.62$ | 75.09 |
| | $k$-NN | $74.00 \pm 0.65$ | $18.17 \pm 0.05$ | $\mathbf{70.27} \pm 3.82$ | $\mathbf{79.74} \pm 2.45$ | 60.54 |
| | kernel | $29.93 \pm 1.33$ | $18.07 \pm 0.21$ | $57.66 \pm 7.75$ | $57.52 \pm 10.42$ | 40.79 |
| | $\partial$Lift | $81.93 \pm 1.11$ | $78.03 \pm 0.91$ | $69.37 \pm 2.55$ | $73.20 \pm 5.62$ | $\mathbf{75.63}$ |
| UniGIN | $k$-hop | $78.73 \pm 0.66$ | $74.47 \pm 1.72$ | $65.77 \pm 9.19$ | $58.82 \pm 5.77$ | 69.44 |
| | $k$-NN | $62.00 \pm 1.08$ | $19.33 \pm 0.48$ | $65.77 \pm 1.27$ | $\mathbf{73.20} \pm 4.03$ | 55.07 |
| | kernel | $40.93 \pm 2.52$ | $18.53 \pm 0.61$ | $58.56 \pm 7.09$ | $51.63 \pm 4.03$ | 42.41 |
| | $\partial$Lift | $\mathbf{84.23} \pm 0.53$ | $\mathbf{77.97} \pm 0.45$ | $63.96 \pm 6.37$ | $63.40 \pm 4.03$ | $\mathbf{72.39}$ |

## 6 CONCLUSION

Topological neural networks (TNNs) are receiving increasing attention in the graph machine learning community. Yet, their effectiveness depends crucially on the choice of graph lifting procedure. Despite its central role, lifting has remained largely unsupervised and task-agnostic, which can lead to the construction of suboptimal topological representations for downstream learning.

To address this limitation, we introduced $\partial$lift, a general-purpose, differentiable lifting framework that is compatible with multiple topological domains. Across a broad set of benchmarks and TNN architectures, $\partial$lift consistently outperformed traditional unsupervised lifting methods, demonstrating the benefit of making the lifting process learnable and task-informed.

**Limitations.** For hypergraph domains, $\partial$lift can create candidate hyper-edges and decide whether to keep them in an embarrassingly parallel fashion — rendering $\partial$lift especially compute-efficient for this domain. However, for cell complexes, we need to compute a cycle basis to elicit candidate cells, which may come at a cubic cost with respect to the number of nodes in the input graph. In this case, we may reduce the number of candidate cells by, for instance, regularizing the $k_v$ variables or shifting their distribution towards zero. Nonetheless, devising more efficient algorithms for candidate identification in hierarchical domains is a clear direction of improvement for future works.

**Future work.** Our method can be extended to other topological domains, such as point clouds, making it applicable to 3D mapping tasks. Additionally, future work could focus on addressing the computational challenges of DiffLift and scaling it to handle larger graphs. Another promising direction is to explore differentiable lifting in dynamic or temporal graphs, where topological

structures evolve over time. Moreover, integrating $\partial$lift with pretraining strategies could yield generalizable topological priors across tasks.

We also believe that formally analyzing the impact of enriching topological structures with learnable liftings on mitigating oversmoothing and oversquashing in TNNs is an interesting research direction. For instance, in scenarios where long and narrow paths connect dense substructures, static liftings are limited to adding cycles within each local region, leaving information between communities to traverse the original bottleneck. In contrast, $\partial$lift can learn to introduce 2-cells that effectively create shortcuts across such bridges, reducing the effective distance between distant nodes and improving information flow. While a full theoretical treatment is left for future work, this adaptive ability provides a plausible mechanism for mitigating oversquashing.

**Ethics Statement.** We do not identify any immediate, direct societal harms from the technical contributions presented in this work. Our method operates on standard, non-sensitive benchmarks and does not require or expose personally identifiable information.

**Reproducibility Statement.** The repository containing the code is available at `https://github.com/JorgeLuizFranco/difflifting`. We provide further details on used datasets and implementation details (e.g., parameter selection) in Appendices D and C.

## ACKNOWLEDGMENTS

We acknowledge the support by the Coordenação de Aperfeiçoamento de Pessoal de Nível Superior (CAPES) (88887.176396/2025-00), Fundação Carlos Chagas Filho de Amparo à Pesquisa do Estado do Rio de Janeiro (FAPERJ) (SEI-260003/020348/2025, SEI-260003/020694/2025) and the Conselho Nacional de Desenvolvimento Científico e Tecnológico (CNPq) (404336/2023-0, 305692/2025-9, 315158/2023-9, 163868/2023-9, 408974/2025-7, 312068/2025-5). We also acknowledge the computational resources provided by the Aalto Science-IT Project.

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

# A  ADDITIONAL BACKGROUND AND FORMULATIONS

## A.1  CHAINS, BOUNDARY OPERATORS, AND CYCLES

Here, we introduce some basic notions in algebraic topology. For simplicity, our exposition considers abstract simplicial complexes (ASCs) equipped with coefficients in the finite field $\mathbb{Z}/2\mathbb{Z} = \{0, 1\}$.

The space of $n$-chains is the vector space of all formal sums of $n$-dimensional simplices of an ASC $K$. Formally, let $n \geq 0$ and $K_{(n)} = \{\sigma \in K : \dim(\sigma) = n\}$ be the $n$-skeleton of $K$. The $n$-chains of $K$ is the set $C_n(K)$ whose elements take the form

$$\sum_{\sigma \in K_{(n)}} \epsilon_\sigma \sigma \tag{8}$$

where for all $\sigma \in K_{(n)}$, $\epsilon_\sigma \in \mathbb{Z}/2\mathbb{Z}$.

Let $c = \sum_{\sigma \in K_{(n)}} \epsilon_\sigma \sigma$ and $c' = \sum_{\sigma \in K_{(n)}} \epsilon'_\sigma \sigma$ be two $n$-chains. The sum of two chains $(c + c')$ and the product of a chain by a scalar $(\lambda c)$ are respectively defined by

$$c + c' = \sum_\sigma (\epsilon_\sigma + \epsilon'_\sigma)\sigma \tag{9}$$

$$\lambda c = \sum_\sigma (\lambda \epsilon_\sigma)\sigma \tag{10}$$

where sums and products are modulo-2.

We define the boundary of a $n$-simplex $\sigma$, denoted by $\partial_n \sigma$ as the sum of its constituents $(n-1)$-simplices, i.e.,

$$\partial_n \sigma = \sum_{\tau \subset \sigma : |\tau| = |\sigma| - 1} \tau \tag{11}$$

This boundary extends linearly to chain spaces. In particular, the boundary operator $\partial_n$ is a linear map $\partial_n : C_n(K) \to C_{n-1}(K)$ defined by

$$\partial_n c = \partial_n \sum_{\sigma \in K_{(n)}} \epsilon_\sigma \sigma = \sum_{\sigma \in K_{(n)}} \epsilon_\sigma \partial_n \sigma. \tag{12}$$

Finally, we can define $n$-cycles. For $n \geq 0$, the $n$-cycles of $K$ is the set $Z_n(K)$ given by the kernel of $\partial_n$, that is

$$Z_n(K) = \{c \in C_n(K) : \partial_n c = 0\}. \tag{13}$$

## A.2 $\partial$lift FOR SIMPLICIAL COMPLEXES

Note that for when $D = 1$ — i.e., when we must decide which edges to add — Steps 2 and 3 of $\partial$lift for cell-complexes naturally result in a simplicial complex. To fully specify $\partial$lift for simplicial complexes, we are left with defining these steps when $D > 1$.

**Case $D \geq 2$: Learning $D$-simplices**

$\Rightarrow$[Step 2] When creating simplices of dimension $D$, we must ensure they respect the hierarchical structure of simplicial complexes. Let $K^\ell$ be the cell complex at the end of iteration $\ell \leq D - 1$. To identify a preliminary set of candidates $\mathcal{C}'$, we run static $D$-clique lifting on $K^1$. For $D > 2$, it is possible that a lower-order clique within some $C \in \mathcal{C}'$ does not belong to $K^{D-1}$. Therefore, we must filter out these elements, defining a refined set of candidates:

$$\mathcal{C} = \{C \in \mathcal{C}' | S \in K^{D-1} \text{ for all } S \subset C\} \tag{14}$$

$\Rightarrow$[Step 3] Similarly to this respective step for cell complexes, we define the probability of accepting $C \in \mathcal{C}$ (i.e., setting $b_C = 1$) applying a DeepSet over the embeddings $\{\!\{z_v\}\!\}_{v \in C}$, subsequently sampling the Bernoulli variables $\{\!\{b_C\}\!\}_{C \in \mathcal{C}}$. The output complex at this iteration is then

$$K^D = K^{D-1} \cup \{C \in \mathcal{C} : b_C = 1\}. \tag{15}$$

We define the features for $D$-simplices using sum projection lifting.

### A.3 $\partial \mathrm{lift}$ FOR COMBINATORIAL COMPLEXES

There are multiple ways to combine cell complexes with hypergraphs to obtain valid combinatorial complexes (CC). Here, we would like to preserve the property that hyperedges exchange messages with nodes via boundary (or lower incidences) neighborhoods. Thus, we propose first running $\partial \mathrm{lift}$ to either cell or simplicial complexes — where ranking functions are given by cell/simplex dimensions. Let $K$ be the resulting complex. Then, we employ (in parallel) $\partial \mathrm{lift}$ to a hypergraph $H$, where edges/hyperedges have rank 1. To ensure a valid combinatorial complex, we prune the sampled hyperedges to include only those that are not supersets of any cell of rank greater than 1 in $K$. Formally, the resulting CC is given by $\{h \in H : \nexists\, \sigma \in K \text{ s.t. } \sigma \subseteq h\} \cup K$.

## B TOPOLOGICAL LIFTINGS

**Clique lifting.** The set of cliques in a graph $G$ is given by $Cl(G) = \{c \subseteq V(G) : u \neq v \in c \implies \{u, v\} \in E(G)\}$, i.e., each element of $Cl(G)$ is a complete subgraph of $G$. The $k$-cliques of $G$ are the elements of $Cl(G)$ of size $k$, for $k > 1$, and we denote them as $Cl_k(G)$. Formally, the $k$-clique lifting operation is given by

$$\mathrm{lift}_{\mathrm{clique},k}(G) = V(G) \cup_{i=2}^{k} Cl_i(G). \tag{16}$$

Note that the inclusion of all cliques of size smaller than $k$ ensures the function returns a valid abstract simplicial complex.

**Cycle lifting.** The idea of cycle lifting is to identify basic cycles in the input graph and use the tuple of vertices in a cycle as a 2-rank cell of the output complex.

Let us consider modulo-2 sum operations for vertices and edges. Also, let $\partial_1$ be the edge boundary map for a graph $G$, i.e., $\partial_1(\{u, v\}) = \{u\} + \{v\}$ for any edge $\{u, v\} \in E(G)$. The cycles of $G$ are $L(G) = \{l \subseteq E(G) : \sum_{e \in l} \partial_1(e) = 0\}$.

A basis for cycles of $G$ is a minimal collection of cycles such that any cycle in $G$ can be written as a sum of cycles in the basis — i.e., the smallest set $B \subseteq L$ such that $\forall l \in L, \exists B' \subseteq B$ with $l = \sum_{b \in B'} b$. The cycle lifting map is

$$\mathrm{lift}_{\mathrm{cycle}}(G) = V(G) \cup E(G) \cup \{V(b) : b \in B(G)\}, \tag{17}$$

where $V(b)$ denotes the set of vertices in the cycle $b$, and $\{V(b) : b \in B(G)\}$ is the set of 2-dim cells.

**DCM.** Battiloro et al. (2024) proposed a novel layer composed of several modules, with the Differentiable Cell Complex Module (DCM) being central to latent topology inference. The DCM first samples the 1-skeleton of the latent cell complex using the $\alpha$-Differentiable Graph Module ($\alpha$-DGM). It then selects polygons—representing higher-order interactions—formed by cycles in the sampled graph using the Polygon Inference Module (PIM). For a detailed description of $\alpha$-DGM and PIM, we refer the reader to Section 3 of Battiloro et al. (2024).

**$k$-hop lifting.** The $k$-hop neighborhood of a node $v \in V(G)$ is defined as

$$N_k[v] = \{u \in V(G) : \mathrm{dist}(u, v) \leq k\}, \tag{18}$$

where $\mathrm{dist}(u, v)$ is the shortest-path distance in the graph $G$, measured by the number of edges in the path.

To construct the $k$-hop hypergraph $H$ from $G$, a hyperedge is formed for each node $v \in V(G)$ based on its $k$-hop neighborhood:

$$\mathrm{lift}_{\mathrm{k\text{-}hop}}(G) = \{N_k[v] : v \in V(G)\}.$$

One can note that when $k = 1$, $k$-hop is equal to neighborhood lifting. The parameter $k$ controls the extent of the neighborhoods included as hyperedges, with larger $k$ values progressively incorporating nodes farther away in terms of shortest-path distance.

**$k$-NN lifting.** $k$-NN lifting constructs hyperedges by identifying the $k$ nearest neighbors based on their node features (feature space). For every node, a separate hyperedge is formed that includes the node itself and its $k$ closest neighbors.

**Kernel lifting.** Kernel lifting is a procedure that constructs hyperedges based on similarity measures derived from kernels over graph nodes. These kernels can be defined in three ways: *(i)* over the graph structure itself, *(ii)* over the node features, or *(iii)* as a composition that jointly incorporates both graph and feature information. For a given reference node $v$, the method computes similarities between $v$ and all other nodes $v^*$ using a kernel function. A hyperedge is then formed by selecting a fixed fraction (typically 0.5) of the nodes that are most similar to $v$ according to the chosen kernel. This process is repeated for each node to construct a set of hyperedges. The kernels can be defined in several forms: over nodes $K_g(v, v^*)$, features $K_x(x, x^*)$, or over nodes and features $C(K(x, x^*, v, v^*))$, where $C$ is a valid composition function. Kernels over features are calculated as standard RBF or exponential kernels (Duvenaud, 2014), whereas kernels over graphs can be calculated as heat or Matérn kernels (Schölkopf and Smola, 2002; Borovitskiy et al., 2021; Nikitin et al., 2022).

## C    IMPLEMENTATION DETAILS

### C.1    MODELS

Our implementation relies mainly on the Pytorch (Paszke et al., 2017) and Pytorch Geometric (Fey and Lenssen, 2019) libraries. For TNN models and static lifting we used TopoX (Hajij et al., 2024) and TopoBench (Telyatnikov et al., 2025).

Regarding the base TNNs, we use the hyperparameters (including learning rate, optimizer, batch size, width, depth, and so on) reported in TopoBench for CWN, CXN, and UniGCNII on NCI1, NCI109, MOLHIV, MUTAG, Proteins, ZINC, Cora and Citeseer (Telyatnikov et al., 2025). Since TopoBench does not report optimal hyperparameters for UniGIN, we use the same used for UniGCNII. While for CIN, we used part of the hyperparameters reported in (Bodnar et al., 2021b) and for the TopoTune, since they do not report the hyperparameters, we used a grid search similar to the one available in their repo.

We note that MOLHIV, Texas and Wisconsin datasets were not present in TopoBench. We use two TNN layers for MOLHIV and one for Texas and Wisconsin with respective learning rates $10^{-2}$, $5 \times 10^{-3}$ and $5 \times 10^{-3}$. For these datasets, we fix the embedding size in 64 for all layers. And for Texas and Wisconsin we used weight decay $5 \times 10^{-6}$.

We are left with the task of optimizing the hyperparameters for the lifting operations ($\partial$lift, $k$-NN, kernel). For $\partial$lift, we consider using both GPS and GIN as backbone GNNs, with embedding dimensions in $\{32, 64, 128\}$, network depth in $\{2, 3\}$, and $k_{\max} = \{3, 5, 7, 9, 11\}$. For $k$-NN lifting we choose $k$ in $\{3, 5, 7, 9\}$. For kernel lifting, we consider equally-spaced temperature values within 0.1 and 9.6, with 0.5 increments.

All models were trained for 200 epochs and with early stopping after 50 epochs without improvement on validation accuracy. We run three independent trials for computing mean and standard deviation of the performance metrics. We select the optimal hyperparameters based on validation accuracy.

### C.2    HARDWARE

For all experiments, we use a cluster with Nvidia V100 GPUs — details regarding the compute infrastructure are omitted for anonymity.

## D    DATASETS

**Graph-level tasks.** The datasets NCI1, NCI109, PROTEINS, and MUTAG are part of the TUDatasets (Kersting et al., 2016) — a dataset collection broadly used for benchmarking GNNs. We also use ZINC-12K and MOLHIV (Hu et al., 2020), popular benchmarks for molecular property prediction. Statistics for each dataset are given in Table 5.

**Node-level tasks.** For node classification, we use four popular benchmarks: Cora, Citeseer (Sen et al., 2008; Shchur et al., 2018), Texas, and Wisconsin (Rozemberczki et al., 2021). Cora and Citeseer are citation networks where nodes represent papers and edges denote citation between them. Node features are given by bag-of-word vectors and node labels comprise the academic topics of

Table 5: Statistics of datasets for graph-level tasks.

| Dataset | #graphs | #classes | Avg #nodes | Avg #edges | Train% | Val% | Test% |
|---------|---------|----------|------------|------------|--------|------|-------|
| NCI1 | 4110 | 2 | 29.87 | 32.30 | 80 | 10 | 10 |
| NCI109 | 4127 | 2 | 29.68 | 32.13 | 80 | 10 | 10 |
| MUTAG | 188 | 2 | 17.93 | 19.79 | 80 | 10 | 10 |
| PROTEINS | 1113 | 2 | 39.06 | 72.82 | 80 | 10 | 10 |
| MOLHIV | 41127 | 2 | 25.5 | 27.5 | Public Split | | |
| ZINC | 12000 | - | 23.16 | 49.83 | Public Split | | |

the underlying articles. Texas and Wisconsin are datasets of webpages from university departments. Nodes represent webpages and edges are hyperlinks between them.

For citation networks, we use the same data split as in (Chen et al., 2018), and for the remaining ones we use the split in (Pei et al., 2020). These are the standard and most used splits. Table 6 provides more details about the datasets.

Table 6: Statistics of datasets for node classification.

| Dataset | #Nodes | #Edges | #Features | #Classes | #Train | #Val | #Test |
|---------|--------|--------|-----------|----------|--------|------|-------|
| Cora | 2708 | 5429 | $1,433$ | 7 | $1,208$ | 500 | $1,000$ |
| Citeseer | 3327 | 4732 | $3,703$ | 6 | $1,827$ | 500 | $1,000$ |
| Texas | 183 | 309 | 1703 | 5 | 87 | 59 | 37 |
| Wisconsin | 251 | 499 | 1703 | 5 | 120 | 80 | 51 |

# E  ADDITIONAL RESULTS

## E.1  RUNTIME AND COMPLEXITY COST

**Wall-clock time.** Table 7 reports per-epoch training and test times (seconds) for the different lifting methods and target neural networks (TNNs) used in our experiments. Overall, $\partial$lift incurs only a modest runtime overhead compared to static liftings while offering the flexibility of task-adaptive topology.

**Memory usage.** Table 8 reports GPU and RAM consumption (GB) for the different architectures and lifting schemes. Reported values are mean $\pm$ standard deviation over runs. Overall, $\partial$lift demonstrates moderate memory requirements across both cellular and hypergraph domains, with GPU usage scaling proportionally to the complexity of the lifted topology.

**Complexity analysis.** Hypergraph $\partial$lift runs in $\mathcal{O}(N^2D+N^2\log k_{\max}+Nk_{\max}D)$, where the $N^2D$ term comes from computing all pairwise node distances in a $D$-dimensional embedding space, the $N^2\log k_{\max}$ term for selecting each node's top-$k$ neighbors, and the $Nk_{\max}D$ term for aggregating across the sampled hyperedges. By contrast, a static kernel-based lifting requires $\mathcal{O}(N^2D + N^3)$ time, while a $k$-hop static lifting scales as $\mathcal{O}(Nk_{\max}\bar{d})$ with $\bar{d}$ the average node degree. Cellular $\partial$lift runs in $\mathcal{O}(N^2D + (E + Nk_{\max})^2 + C\ell D)$, where $(E + Nk_{\max})^2$ captures pairwise adjacency among lifted edges or candidate cells and $C\ell D$ covers the embedding aggregation over $C$ candidate cells of average size $\ell$. A static cycle-basis lifting costs $\mathcal{O}(N(N + E) + C\ell)$, whose dominant component is the cycle-basis computation. In both variants, the modest additional runtime is justified by consistent accuracy gains of approximately 5–10 percentage points compared to static liftings.

**Choice of Hyperparameters.** Although $\partial$lift includes additional learnable components — such as the GNN used for neighborhood scoring, the networks computing acceptance probabilities and adaptive neighborhood sizes, and the maximum neighbor budget $k_{\max}$ — the resulting hyperparameter overhead remains modest. Our search space comprised fewer than 30 configurations overall (Appendix C), and we found that tuning was somewhat stable across datasets. Crucially, the adaptivity introduced by these components leads to consistent performance gains while adding only minor computational and memory costs, as reflected in Tables 7 and 8.

Table 7: Per-epoch training and test times (seconds) for the different lifting methods. Reported values are mean $\pm$ standard deviation over runs.

| Dataset/Phase | CWN Cycle | CWN $\partial$lift | CXN Cycle | CXN $\partial$lift |
|---|---|---|---|---|
| **Cellular Domain** | | | | |
| NCI1 Train | $47.41 \pm 2.59$ | $97.37 \pm 5.02$ | $37.71 \pm 2.31$ | $66.77 \pm 3.00$ |
| NCI1 Test | $1.42 \pm 0.05$ | $4.33 \pm 0.62$ | $1.05 \pm 0.03$ | $2.67 \pm 0.06$ |
| NCI109 Train | $52.11 \pm 2.58$ | $97.05 \pm 3.80$ | $45.76 \pm 0.97$ | $53.50 \pm 2.50$ |
| NCI109 Test | $1.67 \pm 0.02$ | $5.19 \pm 0.41$ | $1.56 \pm 0.02$ | $2.15 \pm 0.08$ |
| MUTAG Train | $2.55 \pm 0.12$ | $4.96 \pm 3.29$ | $1.57 \pm 0.12$ | $3.60 \pm 2.83$ |
| MUTAG Test | $0.09 \pm 0.00$ | $0.23 \pm 0.02$ | $0.05 \pm 0.00$ | $0.12 \pm 0.00$ |
| Proteins Train | $18.07 \pm 1.75$ | $20.12 \pm 2.10$ | $13.63 \pm 2.22$ | $15.18 \pm 1.50$ |
| Proteins Test | $0.65 \pm 0.00$ | $0.82 \pm 0.10$ | $0.46 \pm 0.00$ | $0.72 \pm 0.05$ |
| ZINC Train | $58.24 \pm 3.10$ | $118.45 \pm 6.20$ | $46.18 \pm 2.40$ | $82.30 \pm 4.50$ |
| ZINC Test | $2.15 \pm 0.08$ | $5.92 \pm 0.35$ | $1.68 \pm 0.06$ | $3.84 \pm 0.18$ |
| **Dataset/Phase** | **UniGCNII k-hop** | **UniGCNII $\partial$lift** | **UniGIN k-hop** | **UniGIN $\partial$lift** |
| **Hypergraph Domain** | | | | |
| NCI1 Train | $42.26 \pm 0.46$ | $69.34 \pm 2.55$ | $37.46 \pm 0.31$ | $61.32 \pm 2.83$ |
| NCI1 Test | $1.23 \pm 0.01$ | $1.94 \pm 0.07$ | $0.91 \pm 0.01$ | $2.03 \pm 0.06$ |
| NCI109 Train | $31.66 \pm 2.26$ | $88.22 \pm 3.28$ | $28.62 \pm 2.35$ | $69.61 \pm 3.70$ |
| NCI109 Test | $0.89 \pm 0.05$ | $2.72 \pm 0.03$ | $0.63 \pm 0.02$ | $2.30 \pm 0.05$ |
| MUTAG Train | $1.94 \pm 0.14$ | $5.12 \pm 3.41$ | $2.04 \pm 1.57$ | $3.06 \pm 0.13$ |
| MUTAG Test | $0.06 \pm 0.00$ | $0.14 \pm 0.00$ | $0.05 \pm 0.00$ | $0.09 \pm 0.00$ |
| Proteins Train | $11.93 \pm 0.89$ | $12.45 \pm 1.00$ | $10.12 \pm 2.33$ | $11.50 \pm 1.20$ |
| Proteins Test | $0.36 \pm 0.00$ | $0.60 \pm 0.05$ | $0.24 \pm 0.00$ | $0.50 \pm 0.03$ |
| ZINC Train | $51.80 \pm 1.50$ | $76.70 \pm 3.80$ | $45.30 \pm 1.20$ | $66.40 \pm 3.50$ |
| ZINC Test | $1.58 \pm 0.04$ | $2.68 \pm 0.12$ | $1.28 \pm 0.03$ | $2.85 \pm 0.11$ |

Table 8: Memory usage (GB) for the different lifting methods and architectures. Reported values are mean $\pm$ standard deviation over runs.

| Memory Type | TNN | MUTAG | NCI1 | NCI109 | PROTEINS | ZINC |
|---|---|---|---|---|---|---|
| **GPU Memory (GB)** | | | | | | |
| | UniGCNII k-hop | $0.02\pm0.00$ | $0.02\pm0.00$ | $0.02\pm0.00$ | $0.03\pm0.00$ | $0.02\pm0.00$ |
| | UniGCNII $\partial$lift | $0.02\pm0.00$ | $0.02\pm0.00$ | $0.02\pm0.00$ | $0.05\pm0.01$ | $0.02\pm0.00$ |
| | UniGIN k-hop | $0.02\pm0.00$ | $0.02\pm0.00$ | $0.02\pm0.00$ | $0.02\pm0.00$ | $0.02\pm0.00$ |
| | UniGIN $\partial$lift | $0.02\pm0.00$ | $0.02\pm0.00$ | $0.02\pm0.00$ | $0.04\pm0.00$ | $0.02\pm0.00$ |
| | CWN Cycle | $0.02\pm0.00$ | $0.02\pm0.00$ | $0.02\pm0.00$ | $0.03\pm0.00$ | $0.02\pm0.00$ |
| | CWN $\partial$lift | $0.02\pm0.00$ | $0.02\pm0.00$ | $0.02\pm0.00$ | $0.07\pm0.02$ | $0.02\pm0.00$ |
| | CXN Cycle | $0.02\pm0.00$ | $0.02\pm0.00$ | $0.02\pm0.00$ | $0.03\pm0.00$ | $0.02\pm0.00$ |
| | CXN $\partial$lift | $0.02\pm0.00$ | $0.02\pm0.00$ | $0.02\pm0.00$ | $0.04\pm0.00$ | $0.02\pm0.00$ |
| **RAM (GB)** | | | | | | |
| | UniGCNII k-hop | $1.22\pm0.02$ | $1.31\pm0.01$ | $1.31\pm0.02$ | $1.24\pm0.01$ | $1.34\pm0.00$ |
| | UniGCNII $\partial$lift | $1.45\pm0.02$ | $1.46\pm0.01$ | $1.45\pm0.00$ | $1.46\pm0.01$ | $1.47\pm0.01$ |
| | UniGIN k-hop | $1.21\pm0.01$ | $1.31\pm0.01$ | $1.31\pm0.01$ | $1.24\pm0.01$ | $1.34\pm0.01$ |
| | UniGIN $\partial$lift | $1.42\pm0.01$ | $1.45\pm0.01$ | $1.45\pm0.02$ | $1.42\pm0.00$ | $1.47\pm0.01$ |
| | CWN Cycle | $1.20\pm0.01$ | $1.46\pm0.02$ | $1.45\pm0.00$ | $1.35\pm0.01$ | $1.71\pm0.01$ |
| | CWN $\partial$lift | $1.38\pm0.01$ | $1.42\pm0.01$ | $1.43\pm0.01$ | $1.39\pm0.01$ | $1.42\pm0.01$ |
| | CXN Cycle | $1.21\pm0.00$ | $1.47\pm0.01$ | $1.46\pm0.01$ | $1.37\pm0.02$ | $1.70\pm0.00$ |
| | CXN $\partial$lift | $1.40\pm0.01$ | $1.43\pm0.00$ | $1.44\pm0.00$ | $1.43\pm0.01$ | $1.45\pm0.02$ |

## E.2 DETERMINISTIC VERSION

As we mentioned in the main text, we can derive a deterministic version of $\partial$lift by thresholding probabilities. Table 9 compares $\partial$lift with its deterministic variant trained using `threshold = 0.5`. Note that (random) $\partial$lift outperforms its deterministic counterpart in most cases. Nonetheless, while we fixed the threshold at $0.5$ here, we leave open the possibility that tuning it as a hyper-parameter could yield performance improvements.

Table 9: Comparison between $\partial$lift (random) and its deterministic variant (thresholding at 0.5).

| Domain | TNN | $\partial$lift | NCI1 | NCI109 | ZINC (MAE) | PROTEINS | MUTAG |
|---|---|---|---|---|---|---|---|
| Cellular | CWN | deterministic | $80.62 \pm 0.75$ | $77.64 \pm 0.30$ | $1.33 \pm 5.33$ | $70.54 \pm 0.00$ | $82.46 \pm 2.48$ |
| Cellular | CWN | random | $79.81 \pm 0.40$ | $80.55 \pm 0.50$ | $0.17 \pm 0.00$ | $70.54 \pm 3.34$ | $85.96 \pm 4.96$ |
| Hypergraph | UniGCNII | deterministic | $71.53 \pm 1.92$ | $69.76 \pm 6.03$ | $0.70 \pm 0.01$ | $72.62 \pm 0.42$ | $75.44 \pm 2.48$ |
| Hypergraph | UniGCNII | random | $77.45 \pm 1.88$ | $75.30 \pm 1.10$ | $0.56 \pm 0.03$ | $73.51 \pm 0.84$ | $89.47 \pm 4.30$ |

## E.3 FURTHER COMPARISON AGAINST DCM

**DCM versus $\partial$lift.** DCM (Battiloro et al., 2024) introduces a learnable lifting approach specifically for cell complexes through a two-step procedure: first, the $\alpha$-Differentiable Graph Module ($\alpha$-DGM) learns the 1-skeleton (edges) using $\alpha$-entmax sampling to generate sparse, non-regular graphs; then, the Polygon Inference Module (PIM) samples polygons from induced cycles of the learned graph. DCM uses $\alpha$-entmax for both edge and polygon sampling, and trains end-to-end using auxiliary reward-based losses (Eqs. 11 and 13 in their paper) that encourage edges/polygons involved in correct predictions. The method is evaluated exclusively on node classification tasks (homophilic and heterophilic datasets) and is limited to 2-dimensional cell complexes.

Some key differences between DCM and $\partial$Lift include:

1. *Domain generality*: our framework applies to hypergraphs, simplicial complexes, and combinatorial complexes, not just cell complexes. Extending DCM to hypergraphs is non-trivial because it relies on cycle-based candidate generation, which is inherently tied to the graph structure and does not naturally translate to hyperedge formation;

2. *Sampling mechanism and training*: we use Bernoulli sampling with straight-through estimators rather than $\alpha$-entmax, and critically, we do not require auxiliary reward-based losses — our framework is trained purely with the task loss, making it simpler and better suited for end-to-end learning. Additionally, our hypergraph variant is embarrassingly parallel whereas DCM's two-step procedure is sequential;

3. *Adaptive cell sizes*: we learn distributions over $k_v$ (neighborhood sizes) allowing adaptive hyperedge/cell cardinalities, while DCM's polygon sizes are constrained by the induced cycles in the learned graph;

4. *Evaluation scope*: we assess both node and graph classification across 12 datasets with multiple TNN architectures, demonstrating broader applicability.

**Experiments on point clouds.** We compared $\partial$lift against DCM on point-cloud node classification (graphs with no edges) to provide a direct comparison. Table 10 shows results across four datasets. $\partial$lift combined with different TNNs achieves competitive or superior performance: UniGCNII+$\partial$lift achieves 84.97% on Wisconsin (vs. 71.24% for DCM), and UniGIN+$\partial$lift achieves 83.78% on Texas (vs. 62.16% for DCM), demonstrating substantial gains on heterophilic datasets. On the homophilic dataset Cora, DCM achieves 73.07% compared to 71.47% for the best $\partial$lift variant, showing comparable performance. As expected, the performance in heterophilic datasets are higher in this setting compared to homophilic when referring Table 3 and Table 4. These results demonstrate that $\partial$lift's framework provides competitive performance while offering greater flexibility across topological domains and TNN architectures.

## E.4 COMPARISON AGAINST PROBABILISTIC REWIRING

**IPR-MPNN versus $\partial$lift.** IPR-MPNN (Qian et al., 2024) introduces probabilistic graph rewiring by connecting original graph nodes to a small set of virtual nodes in an end-to-end differentiable

Table 10: Point cloud performance comparison of $\partial$lift across TNNs versus DCM.

| TNN + $\partial$lift | Citeseer | Texas | Wisconsin | Cora |
|---|---|---|---|---|
| UniGCNII | **74.73**$_{\pm 0.12}$ | 81.98$_{\pm 1.27}$ | **84.97**$_{\pm 2.45}$ | 70.83$_{\pm 0.37}$ |
| CWN | 51.80$_{\pm 1.67}$ | 71.17$_{\pm 1.27}$ | 75.82$_{\pm 0.92}$ | 55.90$_{\pm 2.45}$ |
| CXN | 62.03$_{\pm 1.59}$ | 81.08$_{\pm 0.00}$ | 78.43$_{\pm 0.00}$ | 57.43$_{\pm 1.19}$ |
| UniGIN | 72.73$_{\pm 0.61}$ | **83.78**$_{\pm 2.21}$ | 83.66$_{\pm 0.92}$ | 71.47$_{\pm 1.77}$ |
| DCM | 74.40$_{\pm 0.45}$ | 62.16$_{\pm 5.84}$ | 71.24$_{\pm 3.33}$ | **73.07**$_{\pm 0.92}$ |

manner. The method uses an upstream MPNN to compute priors $\theta$ for assigning each original node to $k$ virtual nodes (from $m$ total virtual nodes, where $m \ll n$), sampling assignment matrices via differentiable $k$-subset sampling. A downstream MPNN then operates on the augmented graph with message passing among: (1) original nodes to virtual nodes, (2) among virtual nodes (forming a complete subgraph), and (3) among original nodes.

There are relevant distinctions between $\partial$lift and IPR-MPNN: (i) *Explicit vs. implicit structure*: IPR-MPNN implicitly routes long-range information through virtual nodes, while $\partial$lift explicitly constructs higher-order cells (hyperedges, simplices, polygons) that directly encode multi-way interactions; (ii) *Domain flexibility*: IPR-MPNN operates within the graph domain augmented with virtual nodes, whereas $\partial$lift learns liftings to diverse topological domains (hypergraphs, simplicial complexes, cell complexes, combinatorial complexes); (iii) *Sampling strategy*: both use differentiable $k$-subset sampling, but IPR-MPNN samples node-to-virtual-node assignments while $\partial$lift samples which higher-order cells to include in the lifted structure; (iv) *Computational approach*: IPR-MPNN requires managing virtual node features and specialized message-passing between hierarchies, while $\partial$lift integrates directly with standard TNN architectures designed for each topological domain.

**Results on graph classification.** Table 11 compares IPR-MPNN to $\partial$lift combined with various TNNs on five molecular datasets. To ensure a fair comparison, we evaluated IPR-MPNN using our experimental setup, with the data splits described in Appendix D and a hyperparameter grid similar to the one in Appendix C, with embedding dimensions in $\{32, 64, 128\}$ and network depths in $\{2, 3\}$. All other hyperparameters remained fixed, taken from the configurations in the official IPR-MPNN repository. For instance, on ZINC we do not use edge features.

$\partial$lift achieves substantial improvements on multiple benchmarks: CXN+$\partial$lift reaches 82.08% on NCI1 and 82.57% on NCI109 compared to IPR-MPNN's 77.44% and 77.08%. On ZINC, both CWN+$\partial$lift and CXN+$\partial$lift achieve 0.17 MAE versus IPR-MPNN's 0.39, representing a 56% error reduction. UniGCNII+$\partial$lift obtains the highest MUTAG accuracy at 89.47%. While IPR-MPNN demonstrates the advantages of learnable graph augmentation through virtual nodes, our results show that explicitly learning higher-order topological structures through $\partial$lift can provide complementary benefits, particularly when combined with TNNs designed to exploit these structures.

Table 11: Graph classification performance comparison between IPR-MPNN and $\partial$lift combined with various TNNs

| Method | MUTAG↑ | NCI1↑ | NCI109↑ | PROTEINS↑ | ZINC↓ |
|---|---|---|---|---|---|
| IPR-MPNN | 70.18$_{\pm 2.48}$ | 77.44$_{\pm 1.13}$ | 77.08$_{\pm 0.30}$ | 73.73$_{\pm 1.93}$ | 0.39$_{\pm 0.05}$ |
| CWN + Cycle | 66.67$_{\pm 12.41}$ | 76.93$_{\pm 1.18}$ | 76.71$_{\pm 1.34}$ | 69.05$_{\pm 2.95}$ | 0.46$_{\pm 0.01}$ |
| CWN + $\partial$lift | 85.96$_{\pm 4.96}$ | 79.81$_{\pm 0.40}$ | 80.55$_{\pm 0.50}$ | 70.54$_{\pm 3.34}$ | **0.17**$_{\pm 0.00}$ |
| CXN + Cycle | 61.40$_{\pm 2.48}$ | 72.02$_{\pm 1.69}$ | 75.01$_{\pm 0.62}$ | 70.83$_{\pm 1.52}$ | 0.79$_{\pm 0.02}$ |
| CXN + $\partial$lift | 84.21$_{\pm 4.30}$ | **82.08**$_{\pm 1.50}$ | **82.57**$_{\pm 0.40}$ | 69.94$_{\pm 2.10}$ | **0.17**$_{\pm 0.01}$ |
| UniGCNII + $k$-hop | 61.40$_{\pm 2.48}$ | 72.70$_{\pm 0.52}$ | 72.01$_{\pm 1.55}$ | 72.92$_{\pm 1.11}$ | 0.66$_{\pm 0.02}$ |
| UniGCNII + $\partial$lift | **89.47**$_{\pm 4.30}$ | 77.45$_{\pm 1.88}$ | 75.30$_{\pm 1.10}$ | 73.51$_{\pm 0.84}$ | 0.56$_{\pm 0.03}$ |
| UniGIN + $k$-hop | 64.91$_{\pm 2.48}$ | 65.50$_{\pm 1.99}$ | 66.97$_{\pm 7.25}$ | 71.43$_{\pm 0.73}$ | 1.15$_{\pm 0.01}$ |
| UniGIN + $\partial$lift | 66.67$_{\pm 6.56}$ | 64.88$_{\pm 1.09}$ | 79.74$_{\pm 0.23}$ | 73.81$_{\pm 1.52}$ | 0.92$_{\pm 0.05}$ |

