# OpenReview forum: "Differentiable Lifting for Topological Neural Networks"
_ICLR.cc/2026/Conference — ICLR 2026 Poster_

### Official Review · Reviewer_gFdZ · 2025-10-27

**Soundness:** 2
**Presentation:** 3
**Contribution:** 2
**Rating:** 4
**Confidence:** 4

**Summary:**

This paper addresses the problem of learning liftings, i.e., functions that map graphs to higher-order domains for use in Topological Neural Networks (TNNs). While most existing approaches rely on handcrafted, "static" lifting procedures, this work explores the possibility of learning liftings directly from data in an end-to-end manner. The authors introduce DiffLift, a general framework that can learn to lift attributed graphs to various higher-order topological structures.

The method runs steps:
(i) Generating node embeddings using a standard GNN.
(ii) Constructing candidate higher-order cells or hyperedges.
(iii) Assigning scores to each candidate via a learnable module.
(iv) Sampling candidates using a Bernoulli distribution based on the scores above, using Straight Through Estimators to ensure differentiability.

Two main instantiations of steps (i,ii) are described:
- Graph to Hypergraph lifting: Candidate hyperedges are formed via k-NN search on node embeddings. Their cardinalities are sampled, and scores are computed using a permutation-invariant multiset function.
- Graph to Cell complex lifting: Candidates are derived from cycles in a chosen cycle basis, and scored using a DeepSets-based model.

Experiments are conducted on both graph-level and node-level prediction tasks. When compared with static lifting approaches, DiffLift generally achieves significant improvements. However, the benefits are less consistent on node-wise classification in heterophilic datasets. An additional experiment shows that the choice of GNN backbone (e.g., GPS vs. GIN) in the first stage can noticeably affect performance.

**Strengths:**

[S1] The paper is easy to understand and the narrative quite straightforward.

[S2] The experiments are comprehensive, covering a broad set of use-cases.

**Weaknesses:**

[W1] The relationship between this work and the DCM method is not clearly articulated. The authors cite it, but the concrete technical advantages over DCM are to be discussed. What limitations of DCM are addressed? How? If DiffLift claims more generality across topological domains, the authors should explain why DCM cannot be extended similarly and highlight key methodological differences.

[W2] The proposed learnable lifting mechanism resembles differentiable pooling approaches as well as methods that *learn* to wire virtual nodes in graphs. This paper, e.g., is quite relevant (https://proceedings.neurips.cc/paper_files/paper/2024/file/31df6a082046111e605abfec26ef5ccc-Paper-Conference.pdf). The authors should position their method relative to these lines of work and provide a minimum experimental comparison, in particular, w.r.t. pure hypergraph lifting, where no cycle-like inductive bias is leveraged.

[W3] The requirement to restrict candidate cells to cycles within a chosen basis seems somewhat arbitrary and potentially limiting. If only a subset of those cycles is ultimately selected, it is unclear why a cycle basis is necessary in the first place. Previous works also consider lifting based on any induced or non-induced cycles (see e.g. Bodnar et al., 2021). Is this primarily a computational convenience? A clearer rationale or explanation is needed.

[W4] On graph property prediction benchmarks, the TNN-based baselines very often do not outperform standard GNNs, which raises concerns. Is this expected? It is unclear whether the experimental setup or hyperparameter tuning might be limiting performance.

[W5] For node classification tasks, TNN models are not compared against vanilla GNN baselines. Some lifting methods (e.g., “kernel” on CiteSeer) perform suspiciously poorly without explanation.

**Questions:**

[Q1] What is the difference between using a DeepSet model in the case of Hypergraphs and a Set Function in the case of Cell Complexes in Figure 4?

[Q2] Line 314 – Hajij et al, characterised various lifting together, but other previous uncited works introduced them first?

---

> ### Author Response · Authors · 2025-11-22
> **Part 1/2**
>
> Thank you for your review and feedback. Below, we provide point-by-point responses to your comments.
>
> > [W1] The relationship between this work and the DCM method is not clearly articulated. The authors cite it, but the concrete technical advantages over DCM are to be discussed. What limitations of DCM are addressed? How? If DiffLift claims more generality across topological domains, the authors should explain why DCM cannot be extended similarly and highlight key methodological differences.
>
> Thanks for the opportunity to clarify some key differences between our work and DCM [3]:
> - Generality: While DCM only applies to cell complexes, our frameworks is rather general and applicable to different domains, including hyper-graphs and simplicial complexes;
> - Internal mechanisms for cell generation: while we consider a probabilistic approach using straight-through, DCM uses -entmax and multiple losses. Notably, our framework for hyper-graphs is embarrassingly parallel, while DCM is inherently sequential;
> - Evaluation: Our work assesses lifting operations on multiple benchmarks for node- and graph-level tasks, while DCM only considers node classification.
>
> Regarding our first bullet, we note that it is not clear how we could implement DCM for hypergraphs and combinatorial complexes. To illustrate this, consider the case of hypergraphs. Intuitively, the first step would be to replace cell-cycle lifting with some static hypergraph lifting procedure. However, in the general case, it is not clear how to ensure that candidate hyper-edges are related to generated edges --- to enable learning to accept/reject candidate hyperedges via gradient propagation.
>
> Besides all the differences in their internal mechanisms , another important difference is that candidate cells/hyperedges in DiffLift have adaptive size --- i.e., we are learning . Therefore, we believe that DiffLift is fundamentally different from DCM and represents a meaningful contribution to the TNN community.
>
> > [W2] The proposed learnable lifting mechanism resembles differentiable pooling approaches as well as methods that learn to wire virtual nodes in graphs. This paper, e.g., is quite relevant (https://proceedings.neurips.cc/paper_files/paper/2024/file/31df6a082046111e605abfec26ef5ccc-Paper-Conference.pdf). The authors should position their method relative to these lines of work and provide a minimum experimental comparison, in particular, w.r.t. pure hypergraph lifting, where no cycle-like inductive bias is leveraged.
>
> Thank you for the pointer. This paper [6]  follows a different evaluation setup from ours. For instance, they “perform a 10-Fold Cross-Validation and report the average validation performance” (Appendix D in their paper) — it is unclear how it performs model selection using grid search, and use positional encoders. To ensure a fair comparison, we are currently running additional experiments and we aim to include these results in the revised paper before the end of the rebuttal period.
>
> Disclaimer: the co-author in charge of the experiments is temporarily unavailable (currently on honeymoon 🌝) and will be back soon.
>
> > [W3] The requirement to restrict candidate cells to cycles within a chosen basis seems somewhat arbitrary and potentially limiting […] Previous works also consider lifting based on any induced or non-induced cycles (see e.g. Bodnar et al., 2021). Is this primarily a computational convenience?
>
> Indeed, we used ``cycle_basis`` following [2] to ensure a fair comparison. However, we explicitly acknowledge that it would be possible to use chordless/induced cycles in the manuscript: “We note that we can also employ chordless cycles to define the 2-cells” in line 273.
>
> > [W4] On graph property prediction benchmarks, the TNN-based baselines very often do not outperform standard GNNs, which raises concerns. Is this expected? It is unclear whether the experimental setup or hyperparameter tuning might be limiting performance.
>
> Thanks for raising this important point. While DiffLift drives the TNNs’ performance higher than GNN baselines, the phenomenon you mentioned for TNNs without DiffLifting are consistent with other works in the literature. For instance, TopoBench [2] reports several instances in which standard GNNs outperform TNNs (c.f., Table 6 in their manuscript). That being said, we leave open the possibility that TNN results can be significantly improved if we include standard deep learning paraphernalia in their hyper-parameter grids (e.g., positional encodings, regularization, batch-norm, skip-connections, virtual nodes). On that note, Platonov et al. (2023) [1] showed that vanilla GNNs (e.g., GCN, GAT) can achieve SOTA performance on heterophilic data if carefully tuned. Overall, we believe the area lacks an extensive, rigorous, and fair comparison between classic GNNs and higher-order message passing networks.

---

> ### Author Response · Authors · 2025-11-22
> **Part 2/2**
>
> > [W5] For node classification tasks, TNN models are not compared against vanilla GNN baselines. Some lifting methods (e.g., “kernel” on CiteSeer) perform suspiciously poorly without explanation.
>
> Thanks for the suggestion. We now provide results for GCN/GIN/GAT on node classification, which we will add to the revised paper. In the table below we compare these baselines against TopoTune + DiffLift.
>
> | Dataset | GCN (%) | GAT (%) | GIN (%) | **TopoTune + DiffLift (%)** |
> |---------|---------|---------|---------|----------------------------|
> | **Cora** | 85.64 ± 0.51 | 86.17 ± 0.33 | 85.50 ± 0.54 | **86.82 ± 0.75** |
> | **CiteSeer** | 70.43 ± 0.71 | 73.82 ± 0.45 | 72.20 ± 0.60 | **78.23 ± 1.08** |
> | **Texas** | 58.91 ± 0.76 | 58.38 ± 1.05 | 59.10 ± 0.80 | **72.97 ± 0.00** |
> | **Wisconsin** | 49.14 ± 0.66 | 49.41 ± 0.95 | 48.50 ± 0.70 | **65.36 ± 2.45** |
>
> On homophilic graphs (Cora, CiteSeer), TopoTune + DiffLift yields modest improvements over the best vanilla GNNs. In contrast, on heterophilic graphs (Texas, Wisconsin), the performance gains are substantially larger, indicating that combining learnable liftings with TNNs can meaningfully enhance model effectiveness in more challenging, heterophilic settings.
>
> Regarding the performance of kernel lifting on CiteSeer: It is hard to pin down why kernel lifting performs poorly in this setting. We note that we have contacted one of the authors of kernel lifting  to confirm that our implementation and results are correct. We emphasize that kernel lifting has not been evaluated before in the literature - although it was proposed in ICML TDL Challenge 2024, there was no empirical evaluation. We are the first paper to evaluate kernel lifting for TNNs in a wide range of experiments.
>
> > [Q1] What is the difference between using a DeepSet model in the case of Hypergraphs and a Set Function in the case of Cell Complexes in Figure 4?
>
> Thanks for your question. DeepSet is just a parametrization for set functions, which happens to be a universal approximator for (multi)set functions. In our work, all set functions are implemented either as DeepSets or average/sum pooling. We will clarify this and standardize the notation.
>
> > [Q2] Line 314 – Hajij et al, characterised various lifting together, but other previous uncited works introduced them first?
>
> You are correct! To the best of our knowledge, Bodnar et al. [4, 5] introduced static lifting procedures in the context of topological deep learning. We will properly acknowledge pioneering contributions in the revised manuscript.
>
> References:
>
> [1] A critical look at the evaluation of GNNs under heterophily: Are we really making progress?. ICLR, 2023.
>
> [2] TopoBench: A Framework for Benchmarking Topological Deep Learning. ArXiv, 2024.
>
> [3] From Latent Graph to Latent Topology Inference: Differentiable Cell Complex Module. ICLR, 2024.
>
> [4] Weisfeiler and Lehman Go Topological: Message Passing Simplicial Networks. ICML, 2021.
>
> [5] Weisfeiler and Lehman Go Cellular: CW Networks. NeurIPS, 2021.
>
> [6]  Probabilistic Graph Rewiring via Virtual Nodes. NeurIPS, 2024.

---

> ### Comment · Reviewer_gFdZ · 2025-11-26
> **Response to rebuttal**
>
> Thanks for your rebuttal.
>
> - Regarding [W1]: Thanks for articulating a comparison with DCM. I believe the authors should now make an effort to write an appendix paragraph in a paper revision where they: (i) methodologically introduce DCM and its functioning; (ii) illustrate differences as above; (iii) clarify what they mean by:
>
> > "To illustrate this, consider the case of hypergraphs. Intuitively, the first step would be to replace cell-cycle lifting with some static hypergraph lifting procedure. However, in the general case, it is not clear how to ensure that candidate hyper-edges are related to generated edges --- to enable learning to accept/reject candidate hyperedges via gradient propagation."
>
> - Regarding [W2] Thanks. The authors should now make an effort to write an appendix paragraph in a paper revision where they also (concisely) position their work with respect to this referenced work, other than reporting experimental comparisons.
>
> - Regarding [W4, W5]:
>
> > Overall, we believe the area lacks an extensive, rigorous, and fair comparison between classic GNNs and higher-order message passing networks.
>
> This is, generally, a problematic point if results and conclusions are drawn from these comparisons. How did the authors, for example, ensured fair comparisons in the experimental results reported in replying to W5?
>
> ---
>
> I await the authors' inputs on the above and, potentially, results w.r.t. [W2].
>
> Thanks very much again for the answers.

---

> > ### Author Response · Authors · 2025-12-03
> >
> > Thank you very much for engaging with our rebuttal and for your constructive feedback.
> >
> > > Regarding [W1]: DCM Comparison
> >
> > Thank you for your suggestion. **We have now added a new appendix (Section E.3)**, highlighting key differences between DiffLifting and DCM.
> >
> > Additionally, **we conducted new experiments on point clouds** (graphs with no edges) comparing ∂lift across multiple TNNs versus DCM. The results were added in the paper (Table 10 in Appendix E.3).
> >
> > | **TNN + ∂lift** | **Citeseer** | **Texas** | **Wisconsin** | **Cora** |
> > |-----------------|--------------|-----------|---------------|----------|
> > | UniGCNII | **74.73** ± 0.12 | 81.98 ± 1.27 | **84.97** ± 2.45 | 70.83 ± 0.37 |
> > | CWN | 51.80 ± 1.67 | 71.17 ± 1.27 | 75.82 ± 0.92 | 55.90 ± 2.45 |
> > | CXN | 62.03 ± 1.59 | 81.08 ± 0.00 | 78.43 ± 0.00 | 57.43 ± 1.19 |
> > | UniGIN | 72.73 ± 0.61 | **83.78** ± 2.21 | 83.66 ± 0.92 | 71.47 ± 1.77 |
> > | DCM | 74.40 ± 0.45 | 62.16 ± 5.84 | 71.24 ± 3.33 | **73.07** ± 0.92 |
> >
> >
> > **A note on experimental setup**: We would like to clarify that the dataset splits used by the original DCM implementation are not reproducible. The DCM paper does not describe how the splits were constructed, and their public repository does not provide the information needed to reconstruct them. Since the original splits cannot be reproduced, we evaluated DCM directly in our setup (using the dataset splits described in our Appendix D) to ensure a fair and consistent comparison across all methods.
> >
> > > Regarding [W2]: IPR-MPNN Comparison
> >
> > Thank you for highlighting this relevant related work. **We have added a new section (Appendix E.4) comparing ∂lift against IPR-MPNN**, including i) conceptual and methodological differences between DiffLifting  IPR-MPNN,  ii) empirical comparison on graph classification. In particular, we highlight that IPR-MPNN implicitly routes information through virtual nodes while ∂lift explicitly constructs higher-order cells; IPR-MPNN operates within the graph domain while ∂lift learns liftings to diverse topological domains.
> >
> > **Results on graph classification.** The table below reports the results. $\partial$lift achieves substantial improvements on multiple benchmarks: CXN+$\partial$lift reaches 82.08\% on NCI1 and 82.57\% on NCI109 compared to IPR-MPNN's 77.44\% and 77.08\%. On ZINC, both CWN+$\partial$lift and CXN+$\partial$lift achieve 0.17 MAE versus IPR-MPNN's 0.39, representing a 56\% error reduction. UniGCNII+$\partial$lift obtains the highest MUTAG accuracy at 89.47\%.
> >
> > | **Method** | **MUTAG** ↑ | **NCI1** ↑ | **NCI109** ↑ | **PROTEINS** ↑ | **ZINC** ↓ |
> > |------------|-------------|------------|--------------|----------------|------------|
> > | IPR-MPNN | 70.18 ± 2.48 | 77.44 ± 1.13 | 77.08 ± 0.30 | 73.73 ± 1.93 | 0.39 ± 0.05 |
> > | CWN + Cycle | 66.67 ± 12.41 | 76.93 ± 1.18 | 76.71 ± 1.34 | 69.05 ± 2.95 | 0.46 ± 0.01 |
> > | CWN + ∂lift | 85.96 ± 4.96 | 79.81 ± 0.40 | 80.55 ± 0.50 | 70.54 ± 3.34 | **0.17** ± 0.00 |
> > | CXN + Cycle | 61.40 ± 2.48 | 72.02 ± 1.69 | 75.01 ± 0.62 | 70.83 ± 1.52 | 0.79 ± 0.02 |
> > | CXN + ∂lift | 84.21 ± 4.30 | **82.08** ± 1.50 | **82.57** ± 0.40 | 69.94 ± 2.10 | **0.17** ± 0.01 |
> > | UniGCNII + k-hop | 61.40 ± 2.48 | 72.70 ± 0.52 | 72.01 ± 1.55 | 72.92 ± 1.11 | 0.66 ± 0.02 |
> > | UniGCNII + ∂lift | **89.47** ± 4.30 | 77.45 ± 1.88 | 75.30 ± 1.10 | 73.51 ± 0.84 | 0.56 ± 0.03 |
> > | UniGIN + k-hop | 64.91 ± 2.48 | 65.50 ± 1.99 | 66.97 ± 7.25 | 71.43 ± 0.73 | 1.15 ± 0.01 |
> > | UniGIN + ∂lift | 66.67 ± 6.56 | 64.88 ± 1.09 | 79.74 ± 0.23 | 73.81 ± 1.52 | 0.92 ± 0.05 |
> >
> >
> > **A note on the experimental setup**: We examined IPR-MPNN's public repository to align protocols as closely as possible. Using their exact protocol would require re-running all our experiments under their pipeline. Since our experimental setup was already fully established and reproducible (with clearly defined data splits described in our Appendix, Section "Datasets"), the most coherent approach was to run IPR-MPNN within our setup rather than adapt our entire pipeline to theirs.  We note that our setup employs a standard hold-validation procedure for model selection, avoiding data leakage incurred by IPR-MPNN’s original evaluation. For ZINC specifically, we also replaced the GINE variant with GIN to remove edge features, ensuring a fair comparison across methods.
> >
> >
> > > "How did the authors, for example, ensure fair comparisons in the experimental results reported in replying to W5?"
> >
> > We run all methods using the same train/val/test splits (detailed in Appendix D), the same evaluation metrics, and the same seeds. Regarding GNN’s hyperparameters, we use the same grid as we use for TopoTune. We also refrain from using additional GNN paraphernalia, such as jump knowledge and virtual nodes, in all methods — including both GNNs and TopoTune+DiffLifting.
> >
> > ---
> >
> > Thank you again for your thoughtful and constructive feedback.

---

### Official Review · Reviewer_1MFQ · 2025-10-30

**Soundness:** 3
**Presentation:** 4
**Contribution:** 3
**Rating:** 6
**Confidence:** 3

**Summary:**

The paper proposes a differentiable lifting to create an end-to-end learnable
pipeline for the lifting of graphs into higher order structures to enhance
their expressivity. Their pipeline consists of an initial embedding layer that
maps a combinatorial complex (specialized to graphs in the experiments) into a
point cloud. A cell sampler subsequently samples rank-k cells from this point
cloud and learns whether or not the k-cell should be included in the final
lifted cell complex. Hitherto the standard procedure for this lifting has been
deterministic, based on either cliques or cycles, for instance. The author
argue that the optimal lifting depends heavily on the dataset, motivating the
need for a *learned* lifting in order to mitigate this need for  pre-defined
choices. An experimental suite shows that this method works rather well on a
wide range of standard benchmarks for both node classification as well graph
classification.

**Strengths:**

Overall the paper is well-written and the idea is interesting. In particular
eliminating the choice of lifting provides a more general approach for TNNs to
work on a wider set of graphs. This is an important fact, as multiple works
have already shown that various graph datasets have very different properties
and that it is not always clear what type of networks operates best on each
particular dataset. Hence overcoming this challenge is a good contribution to
the TDL community. The code to reproduce the experiments are provided and is
well-documented, which is great to see.

**Weaknesses:**

While the reviewer views the experiments section good it is a little surprising
that the methods only compare to other TNN methods, whereas other comparison
partners could be interesting as well, such as DiffPool or other methods
proposed by the TDA community and graph learning community in general. As this
method seems to have a higher complexity that other methods, it is in the
opinion important to discuss what may persuade a practitioner to use this
method over say a simpler method if it performs more or less equal.

**Questions:**

- Is there a particular reason the method is restricted to graphs? I could
imagine that, since the idea of TNNs is to operate on higher dimensional
complexes as well, higher dimensional simplicial data also to be interesting.
- Have you considered applying the method to point clouds as well?
- One limitation is that to construct the cell complexes, on has to loop over
the full power set, which get quite intense. It is understandable that one
would only consider 1- and 2-cells, but even that would become rather
prohibitive. E.g. A "tiny" point cloud or mesh of an object quickly contains up
to 2-4k points and this is where your method already could become quite
expensive since even constructing the 1-cells (graph) for a single point clouds
would already have to consider $2048 choose 2 = 2,096,128$ or $4096 choose 2 =
8,386,560$ possibilities and for 2-cells one would have to consider billions of
possibilities. Some remarks could shed some light into these considerations.

---

> ### Author Response · Authors · 2025-11-22
> **Response**
>
> Thank you for your review and feedback. Below, we provide point-by-point responses to your comments.
>
> >  While the reviewer views the experiments section good it is a little surprising that the methods only compare to other TNN methods, whereas other comparison partners could be interesting as well, such as DiffPool or other methods proposed by the TDA community and graph learning community in general. As this method seems to have a higher complexity that other methods, it is in the opinion important to discuss what may persuade a practitioner to use this method over say a simpler method if it performs more or less equal.
>
> We appreciate this suggestion. The table below provides an empirical comparison of DiffLift against DiffPool [1] and RePHINE [4] on the ZINC dataset (no edge features) — baselines results taken from [3]. Notably, DiffLift achieves less than half the error of DiffPool + GCN and RePHINE.
>
> | Model / Setting | ZINC MAE (↓) |
> |-----------------|--------------|
> | MLP | 0.710 ± 0.001 |
> | GCN | 0.469 ± 0.002 |
> | GIN | 0.408 ± 0.008 |
> | **DiffPool + GCN** | **0.466 ± 0.006** |
> | GAT | 0.463 ± 0.002 |
> | MPNN (sum aggregation) | 0.381 ± 0.005 |
> | PNA (with scalers) | 0.320 ± 0.032 |
> | **DiffLift (Ours)** | |
> | CWN + DiffLift | **0.170 ± 0.000** |
> | CXN + DiffLift | **0.170 ± 0.010** |
> |**RePHINE** | **0.411 ± 0.015** |
>
> Nonetheless, we would like to clarify that DiffPool [1] and DiffLift address fundamentally different problems. DiffPool learns soft cluster assignments to coarsen graph representations (reducing 0-cells), while DiffLift learns to enrich graph topology by adding higher-order cells (≥1-cells). These are different operations: DiffPool creates hierarchical pooling from N nodes to M < N super-nodes, whereas DiffLift transforms graphs into hypergraphs or cell complexes by adding hyperedges or 2-cells.
>
> We also want to further clarify that TDA-based methods could be used together with topological message passing (the framework we build DiffLift on), since we kind of use TDA as new features in the learning, thus they are not direct competitors.
>
> >  Is there a particular reason the method is restricted to graphs? I could imagine that, since the idea of TNNs is to operate on higher dimensional complexes as well, higher dimensional simplicial data also to be interesting.
>
> Thanks for your question. We focused on graph prediction problems mainly because they are the current standard in TNN evaluation (c.f., TopoBench [2]). However, we acknowledge that the main principle behind DiffLift applies to higher dimensional complexes as well. For instance, we could use TNNs to compute acceptance probabilities for candidate cells from input simplicial complexes of dimension two, and use them to obtain higher-order topological structures. We believe this can be a compelling research direction if we find interesting benchmarking tasks.
>
> > Have you considered applying the method to point clouds as well?
>
> While we focused on graphs, we highlight that DiffLift applies directly to point clouds (without adaptation), as these can be seen as completely disconnected attributed graphs (no edges). We appreciate your suggestion and we will try our best to provide these additional results in the updated PDF by the end of the rebuttal period.
>
> > One limitation is that to construct the cell complexes, on has to loop over the full power set, which get quite intense. It is understandable that one would only consider 1- and 2-cells, but even that would become rather prohibitive. E.g. A "tiny" point cloud or mesh of an object quickly contains up to 2-4k points and this is where your method already could become quite expensive since even constructing the 1-cells (graph) for a single point clouds would already have to consider  2048 choose 2 = 2,096,128
>  or  4096 choose 2 = 8,386,560
>  possibilities and for 2-cells one would have to consider billions of possibilities. Some remarks could shed some light into these considerations.
>
> Thanks for the opportunity to clarify this matter. When learning $D$-cells, for $D=1$, DiffLift only considers $k_v$ candidate cells for each node $v$ in the original graph. For $D=2$, we elicit candidate cells by running cell-cycle lifting on the graph obtained from the previous step — instead of looping over a power set.
>
>
> References:
>
> [1] Hierarchical Graph Representation Learning with Differentiable Pooling. NeurIPS, 2018.
>
> [2] TopoBench: A Framework for Benchmarking Topological Deep Learning. ArXiv, 2024.
>
> [3] Principal Neighbourhood Aggregation for Graph Nets. NeurIPS, 2020.
>
> [4] Going Beyond Persistent Homology Using Persistent Homology. NeurIPS, 2023.

---

> > ### Comment · Reviewer_1MFQ · 2025-11-25
> >
> > Thank you very much for the additional experiments, I have updated my confidence and will keep my score as is.

---

> > > ### Author Response · Authors · 2025-12-03
> > >
> > > Thank you for your support. Following up on our initial answer, we have updated the manuscript (PDF) to include results on point clouds (node classification) --- cf., Table 10, Appendix E3.

---

### Official Review · Reviewer_AwbL · 2025-11-03

**Soundness:** 3
**Presentation:** 3
**Contribution:** 3
**Rating:** 6
**Confidence:** 4

**Summary:**

This paper introduces DiffLift, a differentiable framework for end-to-end learnable graph lifting to higher-order topological domains, including hypergraphs and cell/simplicial complexes. Instead of relying on static, task-agnostic liftings (e.g., cycles, k-hop neighborhoods), DiffLift learns probabilistic distributions over candidate higher-order cells using node embeddings from a backbone GNN. The method integrates with multiple TNN architectures and supports hierarchical sampling for cellular complexes. Experiments across 12 benchmarks show improvements over static liftings.

**Strengths:**

1. The paper proposes a general formulation for differentiable lifting applicable across multiple domains (hypergraphs, cell complexes, simplicial complexes), which enables task-aware structure learning.

2. Experimental validation across multiple datasets and TNN backbones shows performance gains.

3. The proposed framework is robust enough to be integrated with existing TNN frameworks.

**Weaknesses:**

1. Computational bottlenecks remain for cell complexes, especially in cycle-basis computation (cubic in nodes), limiting scalability to larger graphs.

2. The gains on node classification are less uniform, particularly for homophilic datasets with hypergraph models.

3. The proposed method is Dependent on NN-based k-selection, and probability estimators may introduce additional hyperparameter complexity.

4. There are no explicit theoretical guarantees on representation expressiveness or stability under stochastic lifting.

**Questions:**

1. How does Difflift behave on very large graphs (e.g., OGB-LSC scale), where cycle-basis computation and candidate sampling may be expensive?

2. Could the authors elaborate on how sensitive Difflift is to stochastic sampling noise, the backbone GNN used for embedding generation, and the choice of aggregation function? Did you evaluate alternative backbone models or pooling operators, and do you observe increased variance or instability when weaker embeddings or different set functions are used? Are variance-reduction techniques (e.g., Gumbel-softmax, annealing, deterministic thresholds) helpful?

3. Does Difflift mitigate known issues in deep TNNs, such as over-smoothing or over-squashing, more effectively than static lifting?

4. Can the authors report runtime and memory overhead relative to common static lifting pipelines, particularly in high-dimensional molecular tasks?

---

> ### Author Response · Authors · 2025-11-22
> **Part 1/2**
>
> Thank you very much for the thoughtful and constructive review. Below, we provide point-by-point responses to your comments.
>
> > Computational bottlenecks remain for cell complexes, especially in cycle-basis computation (cubic in nodes), limiting scalability to larger graphs.
>
> We acknowledge that liftings for cell complexes incur higher computational cost than hypergraph-based liftings. Nonetheless, our empirical evaluation in Appendix E (Table 7) shows that the overhead is limited to a 3x slowdown (at least for the datasets we considered). We believe this is well justified by the substantial performance gains over static liftings.
>
> Overall, DiffLift’s computational complexity is comparable to that of other liftings in the literature (see Appendix E). For cell complexes, in fact, the bottleneck is finding cycles — a step that static liftings also require.
>
> Importantly, our method includes mechanisms to mitigate computational costs. First, we control graph sparsity by regularizing $k_v$, penalizing excessively large neighborhoods. Second, we can adjust the acceptance thresholds for candidate cells to further limit complexity. Finally, given the substantial gains achieved by DiffLift, we believe our approach paves the way for fast TNNs equipped with differentiable liftings.
>
> > The gains on node classification are less uniform, particularly for homophilic datasets with hypergraph models.
>
> Thanks for your comment. We agree! While we report results for node classifications for completeness, this phenomenon was already reported by [2] — it is well known that TNNs only offer modest improvements over GNNs on high-homophily benchmarks. Nonetheless, for node classification, DiffLift remains competitive with static liftings and consistently outperforms other lifting methods. That being said, graph-level tasks are typically better suited to evaluate TNNs. In particular, Table 1 demonstrates DiffLift achieves superior performance in 22 out of 24 graph classification experiments.
>
> > The proposed method is Dependent on NN-based k-selection, and probability estimators may introduce additional hyperparameter complexity.
>
> Thanks for the opportunity to elaborate on this matter. There are three sources of additional hype-parameters: the GNN component, the networks computing acceptance probabilities and neighborhood sizes, and the maximum number of neighbors $k_\text{max}$. In practice, our hyperparameter search remained modest — our grid included at most thirty combinations across these parameters (see Appendix C). Moreover, we believe that improved performance naturally entails some additional complexity. In this case, the relatively minor computational overhead is well justified by the substantial gains of DiffLift over other liftings, as shown in Table 1.
>
> > There are no explicit theoretical guarantees on representation expressiveness or stability under stochastic lifting.
>
> We agree that, in general, analyzing the expressive power of different liftings is a promising research direction. However, there are basic questions that are yet to be addressed in the literature. For instance: given a fixed backbone TNN, is kernel-based lifting more expressive than k-hop? Regarding stability, while there are well-established results for 1-WL GNNs [1, 3], we are unaware of stability results for TNNs even for static settings.
>
> > [...] how sensitive Difflift is to stochastic sampling noise, the backbone GNN used for embedding generation, and the choice of aggregation function? Did you evaluate alternative backbone models or pooling operators, and do you observe increased variance or instability when weaker embeddings or different set functions are used? Are variance-reduction techniques (e.g., Gumbel-softmax, annealing, deterministic thresholds) helpful?
>
> In our experience, the selection of backbone GNN plays an important role in DiffLift's performance, as shown in Table 2 of the manuscript. Throughout our experiments, we did not feel the need to optimize the aggregation function and used a simple average for all datasets. Moreover, regarding the stochasticity stemming from sampling, we would like to clarify that we already use the Gumbel-Softmax estimator. Nonetheless, it is still possible to threshold the probability, bypassing sampling completely. In particular, Table 8 in Appendix F compares DiffLift against a deterministic variant using thresholds instead of sampling — even though the latter may incur a significant drop in performance.
>
> | Domain | TNN | Variant | NCI1 | NCI109 | ZINC | MUTAG |
> |--------|-----|---------|------|--------|------|-------|
> | Cellular | CWN | Random | **79.81±0.40** | **80.55±0.50** | **0.17±0.00** | **85.96±4.96** |
> | Cellular | CWN | Deterministic | 80.62±0.75 | 77.64±0.30 | 1.33±5.33 | 82.46±2.48 |
> | Hypergraph | UniGCNII | Random | **77.45±1.88** | **75.30±1.10** | **0.56±0.03** | **89.47±4.30** |
> | Hypergraph | UniGCNII | Deterministic | 71.53±1.92 | 69.76±6.03 | 0.70±0.01 | 75.44±2.48 |

---

> ### Author Response · Authors · 2025-11-22
> **Part 2/2**
>
> > Difflift on very large graphs (e.g., OGB-LSC scale) [...]
>
> Thanks for your question. We would like to note that TNNs (even with static liftings) do not yet scale to graphs with million/billion of nodes or edges (e.g., OGB-LSC MAG240M). For instance, the authors of [2] report OOM (out of memory) for TNNs for simplicial and cell complexes on the Tolokers dataset (11K nodes, 1M edges) using 1TB of system memory, and 8 NVIDIA A30 GPUs, each with 24GB of GPU memory. Indeed, scaling up TNNs has been identified as a major research challenge [4].
>
> > Does Difflift mitigate known issues in deep TNNs, such as over-smoothing or over-squashing, more effectively than static lifting?
>
> This is an excellent question. We believe that DiffLift has the potential to mitigate these issues and provide a few constructions below to substantiate our intuition.
>
> **Over-squashing**: Consider a path graph connecting two dense communities. Static cycle lifting adds cycles only within each community, forcing information between communities to traverse a long path:
> ```
> Community A          Bridge          Community B
>   o--o--o                              o--o--o
>   |\ | /|            o--o--o           |\ | /|
>   | \|/ |                              | \|/ |
>   o--o--o                              o--o--o
> ```
>
> DiffLift can learn to add 2-cells that create "express lanes" across the bridge, reducing the effective distance between communities and alleviating over-squashing. By learning which higher-order structures to include based on the downstream task, DiffLift adapts the topology to improve information flow where static methods cannot.
>
> **Over-smoothing:** Regarding over-smoothing, we are not aware of works investigating the impact of higher-order message passing in accelerating oversmoothing. However, depending on the topology of graphs, the inclusion of higher-order cells may negatively impact the degree of oversmoothing (e.g., if the graph has a hamiltonian cycle and cycle-lifting is used). We note that DiffLifting allows for filtering out candidate cells, potentially mitigating the effect of over-smoothing acceleration.
>
> > Can the authors report runtime and memory overhead relative to common static lifting pipelines, particularly in high-dimensional molecular tasks?
>
> Thanks for your question. We report runtime results below.
>
> | Domain         | TNN      | Lifting   | NCI1 Train     | NCI1 Test     | NCI109 Train   | NCI109 Test   | Proteins Train | Proteins Test |
> | -------------- | -------- | --------- | -------------- | ------------- | -------------- | ------------- | -------------- | ------------- |
> | **Cellular**   | CWN      | Cycle     | `47.41 ± 2.59` | `1.42 ± 0.05` | `52.11 ± 2.58` | `1.67 ± 0.02` | `18.07 ± 1.75` | `0.65 ± 0.00` |
> |                |          | **∂lift** | `97.37 ± 5.02` | `4.33 ± 0.62` | `97.05 ± 3.80` | `5.19 ± 0.41` | `20.12 ± 2.10` | `0.82 ± 0.10` |
> |                | CXN      | Cycle     | `37.71 ± 2.31` | `1.05 ± 0.03` | `45.76 ± 0.97` | `1.56 ± 0.02` | `13.63 ± 2.22` | `0.46 ± 0.00` |
> |                |          | **∂lift** | `66.77 ± 3.00` | `2.67 ± 0.06` | `53.50 ± 2.50` | `2.15 ± 0.08` | `15.18 ± 1.50` | `0.72 ± 0.05` |
> | **Hypergraph** | UniGCNII | k-hop     | `42.26 ± 0.46` | `1.23 ± 0.01` | `31.66 ± 2.26` | `0.89 ± 0.05` | `11.93 ± 0.89` | `0.36 ± 0.00` |
> |                |          | **∂lift** | `69.34 ± 2.55` | `1.94 ± 0.07` | `88.22 ± 3.28` | `2.72 ± 0.03` | `12.45 ± 1.00` | `0.60 ± 0.05` |
> |                | UniGIN   | k-hop     | `37.46 ± 0.31` | `0.91 ± 0.01` | `28.62 ± 2.35` | `0.63 ± 0.02` | `10.12 ± 2.33` | `0.24 ± 0.00` |
> |                |          | **∂lift** | `61.32 ± 2.83` | `2.03 ± 0.06` | `69.61 ± 3.70` | `2.30 ± 0.05` | `11.50 ± 1.20` | `0.50 ± 0.03` |
>
> We are currently conducting efficiency analyses to other datasets, and we will report their outcome (including memory performance) before the end of the rebuttal period in our updated PDF.
>
> References:
>
> [1] Weisfeiler and Lehman Go Cellular: CW Networks. NeurIPS, 2021.
>
> [2] TopoBench: A Framework for Benchmarking Topological Deep Learning. ArXiv, 2024.
>
> [3] Tree Mover's Distance: Bridging Graph Metrics and Stability of Graph Neural Networks. NeurIPS, 2021.
>
> [4] Position: Topological Deep Learning is the New Frontier for Relational Learning. ICML, 2024.

---

> > ### Comment · Reviewer_AwbL · 2025-11-28
> >
> > I thank the authors for their rebuttal, which has addressed some of my concerns.
> >
> > Btw, I recommend adding the discussion on over-squashing and the part on "Dependent on NN-based k-selection" to the paper, which will enhance the paper.

---

> > > ### Author Response · Authors · 2025-12-03
> > >
> > > Thank you for further engaging. We are glad to hear that our initial rebuttal has addressed some of your concerns. Below, we address the remaining point on memory consumption and reply to your latest suggestions.
> > >
> > > **Memory consumption**. Following up on our initial answer, we now provide a detailed report of the memory overhead introduced by our approach. These results are also included in the revised manuscript (Appendix E.1).
> > >
> > > The tables below present a comparison of memory usage (GPU and RAM, in GB) between our method and static liftings. Overall, $\partial$lift exhibits moderate memory requirements across both cellular and hypergraph domains, with GPU consumption scaling proportionally to the complexity of the lifted topology.
> > >
> > > #### GPU Memory (GB)
> > >
> > > | **TNN** | **MUTAG** | **NCI1** | **NCI109** | **PROTEINS** | **ZINC** |
> > > |---------|-----------|----------|------------|--------------|----------|
> > > | UniGCNII k-hop | 0.02 ± 0.00 | 0.02 ± 0.00 | 0.02 ± 0.00 | 0.03 ± 0.00 | 0.02 ± 0.00 |
> > > | UniGCNII ∂lift | 0.02 ± 0.00 | 0.02 ± 0.00 | 0.02 ± 0.00 | 0.05 ± 0.01 | 0.02 ± 0.00 |
> > > | UniGIN k-hop | 0.02 ± 0.00 | 0.02 ± 0.00 | 0.02 ± 0.00 | 0.02 ± 0.00 | 0.02 ± 0.00 |
> > > | UniGIN ∂lift | 0.02 ± 0.00 | 0.02 ± 0.00 | 0.02 ± 0.00 | 0.04 ± 0.00 | 0.02 ± 0.00 |
> > > | CWN Cycle | 0.02 ± 0.00 | 0.02 ± 0.00 | 0.02 ± 0.00 | 0.03 ± 0.00 | 0.02 ± 0.00 |
> > > | CWN ∂lift | 0.02 ± 0.00 | 0.02 ± 0.00 | 0.02 ± 0.00 | 0.07 ± 0.02 | 0.02 ± 0.00 |
> > > | CXN Cycle | 0.02 ± 0.00 | 0.02 ± 0.00 | 0.02 ± 0.00 | 0.03 ± 0.00 | 0.02 ± 0.00 |
> > > | CXN ∂lift | 0.02 ± 0.00 | 0.02 ± 0.00 | 0.02 ± 0.00 | 0.04 ± 0.00 | 0.02 ± 0.00 |
> > >
> > > #### RAM (GB)
> > >
> > > | **TNN** | **MUTAG** | **NCI1** | **NCI109** | **PROTEINS** | **ZINC** |
> > > |---------|-----------|----------|------------|--------------|----------|
> > > | UniGCNII k-hop | 1.22 ± 0.02 | 1.31 ± 0.01 | 1.31 ± 0.02 | 1.24 ± 0.01 | 1.34 ± 0.00 |
> > > | UniGCNII ∂lift | 1.45 ± 0.02 | 1.46 ± 0.01 | 1.45 ± 0.00 | 1.46 ± 0.01 | 1.47 ± 0.01 |
> > > | UniGIN k-hop | 1.21 ± 0.01 | 1.31 ± 0.01 | 1.31 ± 0.01 | 1.24 ± 0.01 | 1.34 ± 0.01 |
> > > | UniGIN ∂lift | 1.42 ± 0.01 | 1.45 ± 0.01 | 1.45 ± 0.02 | 1.42 ± 0.00 | 1.47 ± 0.01 |
> > > | CWN Cycle | 1.20 ± 0.01 | 1.46 ± 0.02 | 1.45 ± 0.00 | 1.35 ± 0.01 | 1.71 ± 0.01 |
> > > | CWN ∂lift | 1.38 ± 0.01 | 1.42 ± 0.01 | 1.43 ± 0.01 | 1.39 ± 0.01 | 1.42 ± 0.01 |
> > > | CXN Cycle | 1.21 ± 0.00 | 1.47 ± 0.01 | 1.46 ± 0.01 | 1.37 ± 0.02 | 1.70 ± 0.00 |
> > > | CXN ∂lift | 1.40 ± 0.01 | 1.43 ± 0.00 | 1.44 ± 0.00 | 1.43 ± 0.01 | 1.45 ± 0.02 |
> > >
> > > >  "I recommend adding the discussion on over-squashing and the part on "Dependent on NN-based k-selection" to the paper, which will enhance the paper."
> > >
> > > We have incorporated these suggestions into the revised PDF. In particular, we have added:
> > > - Appendix E.1: A detailed table of memory usage and a discussion of hyperparameter complexity arising from the NN-based k-selection.
> > > - Conclusion (third paragraph): A clarification on how our method could contribute to mitigating over-squashing.
> > >
> > >
> > > Thank you once again for your valuable feedback.

---

### Author Response · Authors · 2025-12-03
**Summary of updates**

Dear reviewers, ACs, and program chairs,

Thank you for your effort, valuable feedback, and service to the community in these challenging times.

We have addressed all the reviewers' suggestions for additional experiments and discussion in our revised manuscript. A summary of these changes is provided below, and all modifications are highlighted in blue in the updated PDF.

- We properly acknowledged the works of Bodnar et al. (2021a,b) in the related work section (Reviewer ``gFdZ``);
- We added results regarding vanilla GNNs on node classification in Table 3 (Reviewer ``gFdZ``);
- We elaborated on how DiffLift could alleviate oversquashing in TNNs — c.f., Section 6 (Reviewer ``AwbL``);
- We have added an empirical elapsed time and peak memory comparison for DiffLift and other static liftings — c.f., Table 8 in Appendix E.1 (Reviewer ``AwbL``);
- We elaborated on the impact of the DiffLifting’s additional hyperparameters on the computational cost in Appendix E.1 (Reviewer ``AwbL``);
- We provided a discussion on key differences between DCM and DiffLifting in Appendix E.3 (Reviewer ``gFdZ``);
- We provided additional results on point-cloud node classification tasks in Table 11 (Appendix E.3) (Reviewer ``1MFQ``);
- Finally, we presented a conceptual and empirical comparison between DiffLifting and probabilistic graph rewiring (IPR-MPNN) (Reviewer ``gFdZ ``).

It is unfortunate that the leak issue halted further discussion, preventing us from receiving additional feedback from reviewers. While Reviewers ``AwbL`` and ``1MFQ`` reinforced their initial positive ratings, we note that Reviewer ``gFdZ`` did not have the chance to possibly update their score after our last reply. Nonetheless, we believe our rebuttal has addressed their remaining concerns — and we would appreciate it if the AC could consider this in their final decision.

We believe acting on reviewers’ feedback has significantly strengthened this work, and we would like to reiterate our appraisal of their contributions.

Best,

The Authors

---

### Meta-Review · Area_Chair_NEZG · 2026-01-07

**Summary:**

This paper proposes ∂lift (DiffLift), i.e., a general framework for learning graph liftings to hypergraphs and cellular- and simplicial complexes in an end-to-end fashion. The methods leverages learned vertex-level latent representations to identify and parameterize distributions over candidate higher-order cells for inclusion. The paper demonstrates that their approach works well empirically. The reviewers are in general positive about the paper. During the rebuttal, the authors worked hard to provide additional details of additional experiments and discussion.

**Reviewer Concerns:**

I think almost of concerns were addressed by authors.

**Reviewer Scores:**

Two out of three reviewers claim that they keep their scores. Another reviewer might not have a chance to further increase the score.

---

### Decision · Program_Chairs · 2026-01-26

Accept (Poster)